**Article** https://doi.org/10.1038/s41467-022-32493-9

# Tuning moiré excitons and correlated electronic states through layer degree of freedom

**Dongxue Chen** [1,2,10], **Zhen Lian**[2,10], **Xiong Huang** [3,4,10], **Ying Su** [5,10], **Mina Rashetnia**[3], **Li Yan**[2], **Mark Blei**[6], **Takashi Taniguchi** [7], **Kenji Watanabe** [8], **Sefaattin Tongay** [6], **Zenghui Wang** [1] ✉, **Chuanwei Zhang** [5] ✉, **Yong-Tao Cui** [3] ✉ & **Su-Fei Shi** [2,9] ✉

Moiré coupling in transition metal dichalcogenides (TMDCs) superlattices introduces flat minibands that enable strong electronic correlation and fascinating correlated states, and it also modifies the strong Coulomb-interaction-driven excitons and gives rise to moiré excitons. Here, we introduce the layer degree of freedom to the $WSe_2/WS_2$ moiré superlattice by changing $WSe_2$ from monolayer to bilayer and trilayer. We observe systematic changes of optical spectra of the moiré excitons, which directly confirm the highly interfacial nature of moiré coupling at the $WSe_2/WS_2$ interface. In addition, the energy resonances of moiré excitons are strongly modified, with their separation significantly increased in multilayer $WSe_2$/monolayer $WS_2$ moiré superlattice. The additional $WSe_2$ layers also modulate the strong electronic correlation strength, evidenced by the reduced Mott transition temperature with added $WSe_2$ layer(s). The layer dependence of both moiré excitons and correlated electronic states can be well described by our theoretical model. Our study presents a new method to tune the strong electronic correlation and moiré exciton bands in the TMDCs moiré superlattices, ushering in an exciting platform to engineer quantum phenomena stemming from strong correlation and Coulomb interaction.

In a strongly correlated electronic system, Coulomb interactions among electrons dominate over kinetic energy. Recently, two-dimensional (2D) moiré superlattices of van der Waals materials have emerged as a promising platform to study correlated physics and exotic quantum phases in 2D, such as the correlated insulating states

and superconductivity in graphene moiré superlattices[1–14], the Mott insulator state at half band filling and various generalized Wigner crystal states at fractional fillings of the moiré superlattices based on transition metal dichalcogenides (TMDCs)[15–23]. The key to the strong correlation in these systems is the enhanced Coulomb interaction in

[1]Institute of Fundamental and Frontier Sciences, University of Electronic Science and Technology of China, Chengdu, Sichuan, China. [2]Department of Chemical and Biological Engineering, Rensselaer Polytechnic Institute, Troy, NY 12180, USA. [3]Department of Physics and Astronomy, University of California, Riverside, CA 92521, USA. [4]Department of Materials Science and Engineering, University of California, Riverside, CA 92521, USA. [5]Department of Physics, University of Texas at Dallas, Dallas, TX 75083, USA. [6]School for Engineering of Matter, Transport and Energy, Arizona State University, Tempe, AZ 85287, USA. [7]International Center for Materials Nanoarchitectonics, National Institute for Materials Science, 1-1 Namiki, Tsukuba 305-0044, Japan. [8]Research Center for Functional Materials, National Institute for Materials Science, 1-1 Namiki, Tsukuba 305-0044, Japan. [9]Department of Electrical, Computer & Systems Engineering, Rensselaer Polytechnic Institute, Troy, NY 12180, USA. [10]These authors contributed equally: Dongxue Chen, Zhen Lian, Xiong Huang, Ying Su. ✉e-mail: zenghui.wang@uestc.edu.cn; Chuanwei.Zhang@utdallas.edu; yongtao.cui@ucr.edu; shis2@rpi.edu

2D and greatly reduced kinetic energy in the flat moiré minibands. In TMDC-based moiré superlattices, the combination of large effective mass and strong moiré coupling renders the easier formation of flat bands and stronger electronic correlation, compared with graphene moiré superlattices. For example, the 0- or 60-degree angle-aligned WSe$_2$/WS$_2$ exhibits Mott insulating states with transition temperatures exceeding 150 K[18,19], the highest among all 2D moiré systems studied so far. It also hosts various correlated insulating states at fractional fillings of the moiré lattice[18,19], indicating strong and long-range electron interactions.

Meanwhile, the strong Coulomb interaction in 2D also leads to tightly bound excitons with large binding energy in TMDCs[24–27]. The moiré coupling in the TMDC moiré superlattices is expected to generate excitonic flat minibands[28], beyond the single-particle electronic flat bands in the conduction and valence bands. Recently, the moiré excitons have been reported in the angle-aligned WSe$_2$/WS$_2$ heterojunction[17,18], in which correlated insulating states also occur[15,17–19]. The excitonic flat band is promising for realizing topological exciton states and correlated exciton Hubbard model[28,29], ushering in exciting opportunities for engineering correlated quantum states. However, there are key questions that remained to be addressed. For example, how is the moiré coupling's extension in the out-of-plane direction? How could one systematically tune both the electronic flat bands and moiré exciton bands in the TMDCs moiré superlattice?

In this work, we investigate these questions utilizing the layer degree of freedom, inspired by the layer-layer coupling in TMDCs that leads to the abrupt direct-to-indirect bandgap transition from monolayer to bilayer TMDCs[30,31]. We demonstrate a general approach tuning both electronic and moiré exciton bands by increasing the layer

number of WSe$_2$ in the angle-aligned WSe$_2$/WS$_2$ heterojunction. As the layer number of WSe$_2$ varies from monolayer (1 L) to bilayer (2 L) and trilayer (3 L), the optical spectra of the moiré exciton change systematically in a way that suggests the moiré coupling is highly interfacial, strongly confined at the WSe$_2$/WS$_2$ interface and barely affects the next neighboring WSe$_2$ layer(s). However, the added WSe$_2$ layer(s) could modify moiré excitons in the WSe$_2$ layer interfacing WS$_2$, resulting in a significant increase in the resonance energy separations between moiré excitons. This observation can be well described by a phenomenological model. Our work, to our best knowledge, reports the first sensitive tuning of moiré excitons via layer degree of freedom.

The correlated electronic structure is also sensitive to the number of layers in WSe$_2$. The Mott insulator state at the filling of one hole per moiré unit cell ($n = -1$) is found to have a transition temperature decreased from 180 K in 1 L/1 L WSe$_2$/WS$_2$ to 120 K in 2 L/1 L WSe$_2$/WS$_2$ and 60 K in 3 L/1 L WSe$_2$/WS$_2$. The correlated states at fractional fillings (fractional charge per moiré supercell) are significantly quenched in the 3 L/1 L WSe$_2$/WS$_2$ heterojunction. The reduced Mott transition temperature, however, is still significantly higher than that of graphene moiré superlattices (~4 K[1]). Our study, therefore, also demonstrates a new knob to tune the strong electron correlation in TMDC moiré superlattices that can be further exploited for engineering new correlated quantum states.

## Results and discussion

The back-gated angle-aligned WSe$_2$/WS$_2$ heterojunction device is schematically shown in Fig. 1a, which includes three different regions in the same device: 1 L/1 L WSe$_2$/WS$_2$, 2 L/1 L WSe$_2$/WS$_2$, and 3 L/1 L WSe$_2$/WS$_2$. The device was constructed through a dry pickup method described previously[13] (also see Methods), and the heterojunctions are

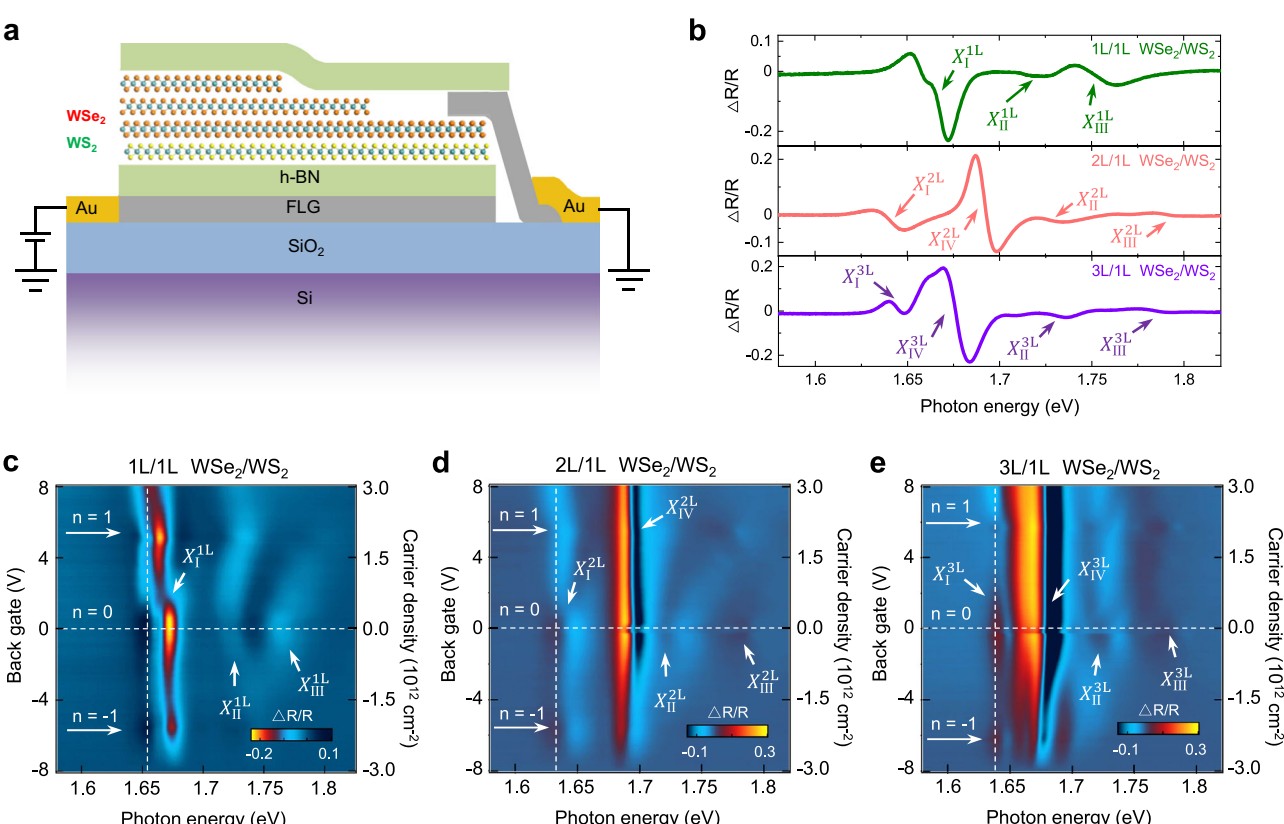

**Fig. 1 | Angle-aligned multilayer WSe$_2$/monolayer WS$_2$ moiré superlattice. a** Schematic of the multilayer WSe$_2$/monolayer WS$_2$ heterojunction devices, with the heterojunctions encapsulated with flakes of BN on both sides. The few-layer-graphene (FLG) works as the back gate electrode. **b** Differential reflectance spectra of different regions at zero gate voltage. **c–e** are the differential reflectance spectra as a function of the gate voltage (density of carriers) at the region of 1 L/1 L, 2 L/1 L, and 3 L/1 L WSe$_2$/WS$_2$. All data were taken at 4.5 K.

encapsulated with flakes of boron nitride (BN) and gated through few-layer graphene flake working as the back gate electrode. WSe$_2$ and WS$_2$ have a lattice mismatch of ~4%, resulting in a moiré superlattice with a periodicity of ~8 nm[32,33] when they are angle-aligned (0- or 60-degree twisted). Having the three regions (1 L/1 L WSe$_2$/WS$_2$, 2 L/1 L WSe$_2$/WS$_2$, and 3 L/1 L WSe$_2$/WS$_2$) in the same device is advantageous as these three regions have the same twist angle since the WSe$_2$ for all these three regions are from the same flake. As a result, the moiré lattice constant in these three regions are about the same, and we can compare our measurements from these different regions directly.

The 1 L/1 L WSe$_2$/WS$_2$ heterojunction has a type-II band alignment, with the conduction band minimum located in the WS$_2$ layer and the valance band maximum in the WSe$_2$ layer[15]. Strong moiré coupling leads to a band folding in the mini-Brillouin zone and generates moiré exciton bands[32], which will split the A exciton resonance of WSe$_2$ into three moiré exciton peaks, as demonstrated in the previous experiments[17,32]. Here we measure the reflectance spectra in the three different regions of the WSe$_2$/WS$_2$ heterojunction as a function of the gate voltage, with results shown in Fig. 1c–e. There are two major differences between the moiré exciton spectra from the 1 L/1 L WSe$_2$/WS$_2$ and multilayer WSe$_2$/monolayer WS$_2$ (2 L/1 L WSe$_2$/WS$_2$ or 3 L/1 L WSe$_2$/WS$_2$) heterojunctions. First, it is evident that near the charge-neutral region (gate voltage ~0 V), there are three moiré exciton resonances in the 1 L/1 L WSe$_2$/WS$_2$ region (Fig. 1c) but four in both 2 L/1 L (Fig. 1d) and 3 L/1 L (Fig. 1e) WSe$_2$/WS$_2$ regions. Second, the moiré exciton energy difference between the lowest and highest energy moiré excitons increases significantly in both 2 L/1 L (Fig. 1d) and 3 L/1 L (Fig. 1e) WSe$_2$/WS$_2$ regions. These observations are better illustrated in Fig. 1b, which plots the differential reflectance spectra at the gate voltage of 0 V for the three different regions (line cuts at zero gate voltage in Fig. 1c–e. For the 1 L/1 L WSe$_2$/WS$_2$ region, we observe three moiré exciton peaks at ~1.662 eV ($X_I^{1L}$), 1.715 eV ($X_{II}^{1L}$), and 1.753 eV ($X_{III}^{1L}$), consistent with the previous reports[32]. However, in the 2 L/1 L WSe$_2$/WS$_2$ region, one additional exciton resonance emerges, adding up to a total of four major excitons peaked at ~1.642 eV ($X_I^{2L}$), ~1.693 eV ($X_{IV}^{2L}$), 1.728 eV ($X_{II}^{2L}$), and 1.793 eV ($X_{III}^{2L}$). In the 3 L/1 L WSe$_2$/WS$_2$ region, there are also four major exciton resonances at ~1.645 eV ($X_I^{3L}$), ~1.677 eV ($X_{IV}^{3L}$), 1.730 eV ($X_{II}^{3L}$), and 1.785 eV ($X_{III}^{3L}$). The largest moiré exciton energy difference, defined as the energy difference between $X_{III}$ and $X_I$, is ~90 meV for the 1 L/1 L WSe$_2$/WS$_2$ region but ~150 meV for 1 L/2 L WSe$_2$/WS$_2$ region and ~140 meV for 1 L/3 L WSe$_2$/WS$_2$ region, an increase of more than 50%. Similar behaviors have been observed for all the devices we have studied (details in Supplementary Note 5).

Our observations suggest that the moiré potential is highly localized at the WSe$_2$/WS$_2$ interface and has a limited extension along the out-of-plane direction. As a result, the moiré coupling only significantly modifies the first WSe$_2$ layer in contact with the monolayer WS$_2$. The newly developed exciton resonances in the 2 L/1 L and 3 L/1 L WSe$_2$/WS$_2$ heterojunctions ($X_{IV}^{2L}$ and $X_{IV}^{3L}$), therefore, arise from the barely modified intralayer A exciton in the upper WSe$_2$ layers away from the interface (also see Supplementary Note 1). Our interpretation is supported by the fact that the energies of $X_{IV}^{2L}$ (1.693 eV) and $X_{IV}^{3L}$ (1.677 eV) are close to the intralayer A exciton energy of monolayer WSe$_2$ (~1.70 eV), and it is further corroborated by the stronger reflectance intensity from the new moiré exciton in 3 L/1 L WSe$_2$/WS$_2$ ($X_{IV}^{3L}$) compared with that in 2 L/1 L WSe$_2$/WS$_2$ ($X_{IV}^{2L}$). Moreover, there is a redshift in the moiré exciton resonance $X_I$ and blueshifts in $X_{II}$ and $X_{III}$ in both 2 L/1 L and 3 L/1 L WSe$_2$/WS$_2$ compared with those in 1 L/1 L WSe$_2$/WS$_2$ (Fig. 1b). And the shift of $X_{II}$ and $X_{III}$ is more significant in magnitude than that of $X_I$.

Our results can be understood with a phenomenological model (details in Supplementary Note 2), considering the moiré excitons in the first WSe$_2$ layer interacting with a exciton state in the added WSe$_2$ layer (s) that has the resonance energy between $X_I$ and $X_{II}$. The resulting level repulsion naturally explains the redshift of $X_I$ and blue shift of $X_{II}$ and $X_{III}$ in the 2 L/1 L (3 L/1 L) WSe$_2$/WS$_2$ compared with 1 L/1 L WSe$_2$/WS$_2$. To

understand the phenomenological model, we propose a possible microscopic mechanism by considering the hybridization between moiré excitons and interlayer-like hybrid exciton ($iX$) in multilayer WSe$_2$/1 L WS$_2$ (details in Supplementary Information Note 2). The hybridization can increase the energy separation between moiré excitons and is enabled by the moiré-potential-induced Umklapp scattering[34–36]. The interlayer-like hybrid exciton arises from the interlayer tunneling in multilayer WSe$_2$ that hybridizes the valence bands in different layers[37]. However, it has much weaker oscillator strength than the intralayer-like hybrid exciton and cannot be resolved in the experiment. In the absence of hybridization between different excitonic states, the energy dispersion of bare intralayer A excitons in 1 L WSe$_2$ and moiré excitons in 1 L/1 L WSe$_2$/WS$_2$ are shown in Fig. 2a, b, respectively. Here we fold the energy bands of A excitons into the mini-Brillouin zone to compare directly with that of moiré excitons. The bright A exciton state at the mini-Brillouin zone center (which is denoted as $\gamma$ point as shown in the inset of Fig. 2a) is marked as $X_A$ in Fig. 2a, and the bright moiré exciton states are marked as $X_I^{1L}$, $X_{II}^{1L}$, and $X_{III}^{1L}$ in Fig. 2b. The optical absorption spectrum of moiré excitons in 1 L/1 L WSe$_2$/WS$_2$ is shown in Fig. 2c, with the three resonances corresponding to the three bright moiré exciton states. For the absorption spectrum of 2 L/1 L WSe$_2$/Ws$_2$ in Fig. 2d, we introduce the hybridization between moiré excitons and interlayer-like hybrid excitons (see Supplementary Information Note 2). The hybridization induces a redshift in $X_I^{2L}$ and blueshifts in $X_{II}^{2L}$ and $X_{III}^{2L}$ compared with those of 1 L/1 L WSe$_2$/Ws$_2$ in Fig. 2c. The larger shift in the magnitude of $X_{II}^{2L}$ and $X_{III}^{2L}$ indicates stronger hybridization with the interlayer-like hybrid exciton, which is consistent with the proposed mechanism (see Supplementary Information Note 2).

Moreover, the intralayer-like hybrid exciton in the second WSe$_2$ layer leads to another resonance $X_{IV}^{2L}$ between $X_I^{2L}$ and $X_{II}^{2L}$, as shown in Fig. 2d, which is consistent with our experimental observation (Fig. 1b, d). In 3 L/1 L WSe$_2$/WS$_2$, additional hybrid excitons can be induced by the interlayer tunneling between valence bands in the 2$^{nd}$ and 3$^{rd}$ WSe$_2$ layers. In this case, the additional hybrid excitons away from the WSe$_2$/WS$_2$ interface do not affect the moiré excitons. Therefore, the moiré excitons in 3 L/1 L WSe$_2$/WS$_2$ are nearly identical to those in 2 L/1 L WSe$_2$/WS$_2$ (Fig. 2d), also consistent with our experimental results (Fig. 1b). On the other hand, the additional hybrid excitons from upper WSe$_2$ layers will contribute to the resonant peak $X_{IV}^{3L}$ which should be consisted of two sub-resonances. This is also consistent with our experimental data, as $X_{IV}^{3L}$ (Fig. 1e) is broader than $X_{IV}^{2L}$ (Fig. 1d). Interestingly, although these two resonances in $X_{IV}^{3L}$ cannot be resolved at the charge-neutral region, likely due to linewidth broadening, they can be revealed in the p-doping region (Fig. 1e). The exact mechanism will be investigated in the future.

The moiré excitons in the three regions show distinct gate dependence, which also confirms the interfacial nature of the moiré coupling in the WSe$_2$/WS$_2$ superlattice. Our theoretical model, which considers the interfacial nature of the moiré coupling, shows that the valence bands due to the added layers are higher in energy than the moiré electronic flat band from the WSe$_2$/WS$_2$ interface, as shown in Fig. 3a–c (detailed calculations in Supplementary Information Note 3). When carriers are added to the 1 L/1 L WSe$_2$/WS$_2$ heterostructure, they will fill the first moiré valence band in the WSe$_2$ layer and the first moiré conduction band in the WS$_2$ layer. The first flat moiré miniband has a strong electron correlation due to their narrow bandwidth, and at the half-filling states (one electron/hole per moiré unit cell, $n$ = +1 or −1), Mott insulator states will occur, as demonstrated in several recent experiments[15–20]. The optical reflectance spectra are expected to be modulated by these correlated states. In the 1 L/1 L WSe$_2$/WS$_2$ region, all three excitons are modulated, with $X_I^{1L}$ being the most obvious one (Fig. 1c). In the 2 L/1 L and 3 L/1 L WSe$_2$/WS$_2$ regions, the excitons at the lowest energy ($X_I^{2L}$ and $X_I^{3L}$) are also strongly modulated (Fig. 1d, e). Figure 3d–f plots the gate voltage dependence of the lowest energy

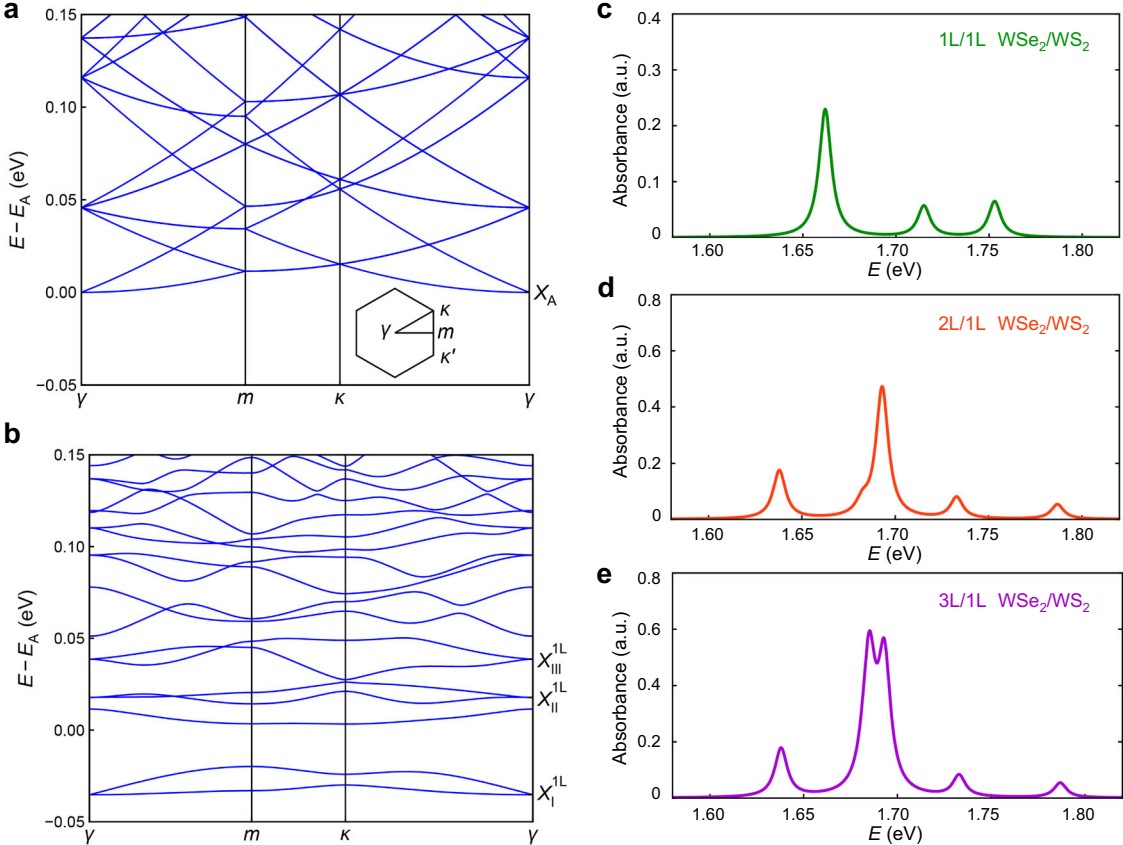

**Fig. 2 | Theoretical simulation of moiré excitons. a, b** are the energy bands of bare intralayer A excitons in 1 L WSe$_2$ and moiré excitons in 1L/1L WSe$_2$/WS$_2$, respectively. **a** The A exciton bands are folded into the mini-Brillouin zone of the moiré superlattice to compare directly with that of moiré excitons in **b**. Inset of **a** shows the schematic of the mini-Brillouin zone and the label of high symmetry points. The WSe$_2$ bright intralayer A exciton state is marked by $X_A$ in **a**. The three bright Moiré exciton states are marked by $X_I^{1L}$, $X_{II}^{1L}$, and $X_{III}^{1L}$ in **b**. Here, we set the energy $E_A$ of $X_A$ as the energy reference. **c–e** Optical absorption spectra in 1L/1L, 2L/1L, and 3L/1L WSe$_2$/WS$_2$. The interlayer hybridization between moiré exciton and intralayer A exciton is considered in **d** and **e**, with details elaborated in Supplementary Information Note 2.

moiré exciton for the three different regions ($X_I^{1L}$, $X_I^{2L}$ and $X_I^{3L}$), which clearly shows intensity modulations at $n = −1$ and $+1$. On the other hand, the additional excitons in 2 L/1 L ($X_{IV}^{2L}$) and 3 L/1 L ($X_{IV}^{3L}$) WSe$_2$/WS$_2$ regions are barely affected by the formation of the Mott states at $n = ±1$ (Fig. 1d, e). These behaviors can also be explained by the interfacial nature of the moiré coupling, which confines the correlated electrons at the interface of WSe$_2$/WS$_2$. The modulation of the moiré excitons at the $n = ±1$ is likely due to the dielectric constant change and gap opening associated with the Mott insulator states. Due to the small radius of the strongly bound exciton[24], only the moiré excitons in the first WSe$_2$ layer immediately interfacing with the WS$_2$ monolayer can sensitively detect the dielectric constant change at the interface. In the 2 L/1 L and 3 L/1 L regions, the additional excitons originated from intralayer excitons localized in the added layers, are thus barely affected.

To better investigate the tuning of the electron correlation by the layer degree of freedom, we perform microwave impedance microscopy (MIM) measurements to study the correlated insulating states in the three different heterostructure regions (Fig. 4a). MIM probes the local conductivity of the sample and has been successfully employed to reveal a rich structure of correlated insulating states in the angle-aligned 1 L/1 L WSe$_2$/WS$_2$ device[8]. In the multilayer WSe$_2$/1 L WS$_2$ device, we primarily focus on the features on the hole side, as the holes reside in the WSe$_2$ layer due to the type-II alignment, and we introduce the layer degree of freedom by modulating the layer number of WSe$_2$. At temperature $T = 10$ K, the MIM spectra in both 1 L/1 L and 2 L/1 L WSe$_2$/WS$_2$ regions show similar pronounced features at various fillings, including

the Mott insulator states at $n = −1$, the generalized Wigner crystal states at fractional fillings of $n = −1/3$ & $−2/3$, $−1/2$, $−1/4$ & $−3/4$, etc. The 3 L/1 L WSe$_2$/WS$_2$ data show fewer and less pronounced dips: other than the Mott insulator state at $n = −1$, only two fractional fillings $n = −1/3$ and $−1/2$ can be resolved. There is also a small difference in the twist angle in the 3 L/1 L WSe$_2$/WS$_2$ region (-1.3°) compared to that in the 1 L/1 L and 2 L/1 L regions (-0.9°), which results in different gate voltage positions for these insulating states in the 3 L/1 L WSe$_2$/WS$_2$ region (details in method). Since the formation of the correlated insulating states at fractional fillings depends on long-range Coulomb interaction among electrons in neighboring moiré unit cells, our results suggest that the inter-site electron interaction strength is weaker in 3 L/1 L than in 1 L/1 L or 2 L/1 L WSe$_2$/WS$_2$. The difference in the on-site interaction, corresponding to the Mott insulator state at $n = −1$, can be further revealed in its temperature dependence. As shown in Fig. 4b–d, as the temperature is raised, the features at fractional fillings disappear at ~30 K in both 1 L/1 L and 2 L/1 L WSe$_2$/WS$_2$ regions and at ~15 K in the 3 L/1 L WSe$_2$/WS$_2$ region. The Mott insulator state at $n = −1$ survives at much higher temperatures in 1 L/1 L WSe$_2$/WS$_2$, persisting to above 180 K, the highest Mott transition temperature reported in all 2D moiré superlattice structures so far. In the 2 L/1 L WSe$_2$/WS$_2$ region, the Mott transition temperature is ~120 K, while it is much lower, ~60 K, in the 3 L/1 L region. As the correlation strength is determined by the ratio of the Coulomb interaction to the kinetic energy, the reduction of electron correlation strength from 1 L/1 L or 2 L/1 L WSe$_2$/WS$_2$ is likely due to the increased dielectric screening from the added WSe$_2$, which reduces the Coulomb

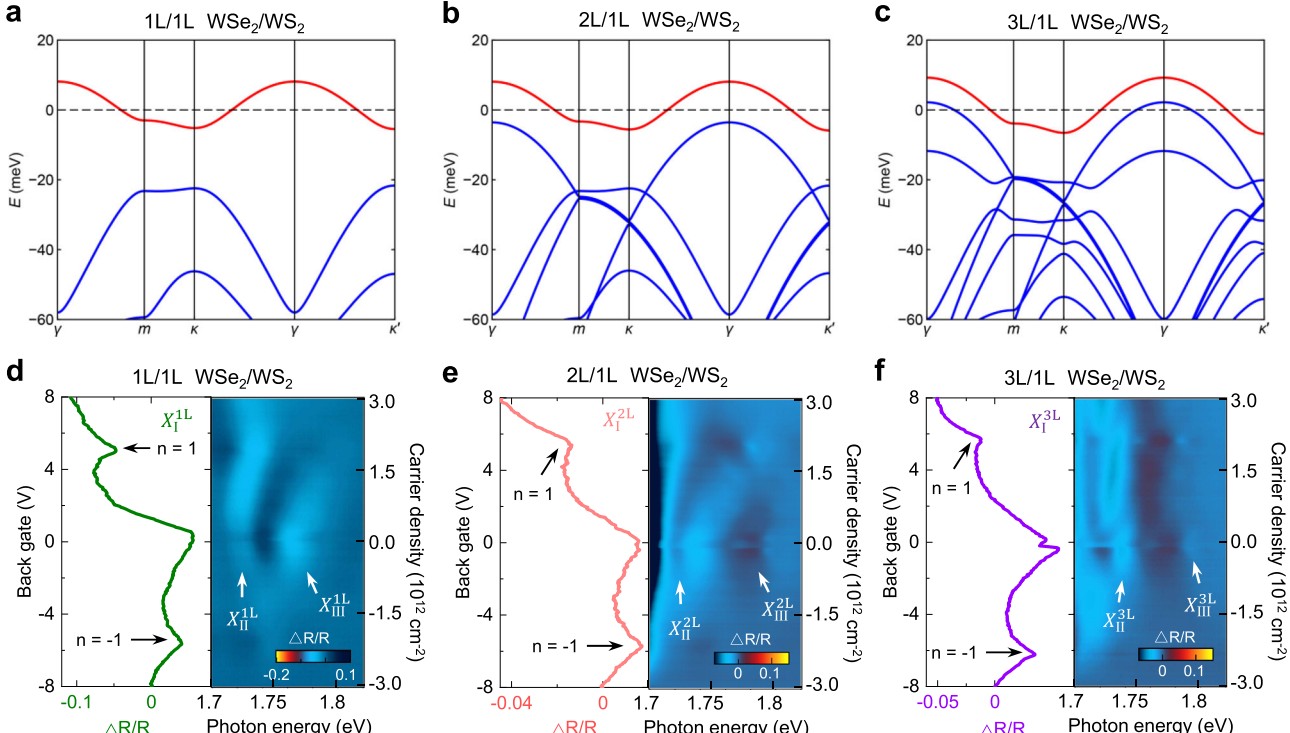

**Fig. 3 | Layer dependence of the electronic flat miniband for WSe₂/WS₂ moiré superlattices. a**–**c** are calculated electronic bandstructure of the valence band in 1 L/1 L WSe₂/WS₂, 2 L/1 L WSe₂/WS₂, 3 L/1 L WSe₂/WS₂, respectively (details in Supplementary Information Note 3), with the moiré flat band from the 1st layer WSe₂ labeled in red. **d**–**f** are differential reflectance intensity of the lowest energy moiré exciton ($X_1$) as a function of the back gate voltage (carrier density) in Fig. 1c–e, respectively (vertical dashed line cuts), with the 2D color plots showing the enhanced reflectance spectra near the moiré excitons $X_{II}$ and $X_{III}$.

interaction at the interface. However, the further reduced correlation strength in the 3 L/1 L WSe₂/WS₂ is facilitated by the additional increase of kinetic energy, which arises from the increased bandwidth of the flat band, according to our calculation shown in Supplementary Information Fig. 2. We emphasize here that even the reduced electron correlation in the 2 L/1 L and 3 L/1 L WSe₂/WS₂ is still significantly stronger than that in graphene moiré systems, which has a Mott transition temperature ~4 K[1]. As a result, the layer degree of freedom can be utilized for engineering new correlated states.

In summary, we have demonstrated a new moiré superlattice system based on multilayer TMDC heterojunctions. The added layers host additional intralayer excitons that interact with the moiré excitons residing at the moiré interface, and they can further modify the correlation strength of the correlated states. Considering the layer-valley-spin locking in TMDC[38], these new TMDCs moiré superlattices provide an exciting platform to investigate emerging correlated valley and spin physics.

## Methods
### Heterostructure device fabrication
We use a dry pickup method[20,39] to fabricate the WSe₂/WS₂ heterostructures. We exfoliate monolayer WS₂, multilayer WSe₂, graphite, and BN layers on silicon substrate with a 285 nm thermal oxide layer. For angle-aligned heterostructures, we choose exfoliated WS₂ and WSe₂ layers with sharp edges, whose crystal axes are further confirmed by second harmonic generation measurements. We then mount the SiO₂/Si substrate on a rotational stage and clamp the glass slide with thin flakes to another three-dimensional (3D) stage. We adjust the 3D stage to control the distance between substrates and thin flakes, and we sequentially pickup different layers onto the pre-patterned Au electrodes on SiO₂/Si substrates. We fine adjust the angle of the rotational stage (accuracy of 0.02°) under a microscope objective to stack the WSe₂/WS₂ heterojunction, ensuring a near-zero twist angle

between the two flakes. The final constructed device is annealed at 130 °C for 12 hours in a vacuum chamber. The pre-patterned Au contact electrodes are fabricated through standard electron-beam lithography and e-beam evaporation processes (see Supplementary Information Fig. 3 for the optical microscope image of the device used in the main text). More sample characterization details can be found in Supplementary Note 4.

### Optical spectroscopy measurements
To perform differential reflectance contrast measurement, the samples were mounted in a helium flow-controlled cryostat with a quartz optical window and electrical feedthroughs. A super-continuum laser (YSL Photonics) was used as the white light source. The laser was focused onto the sample with a ×50 objective (the typical laser spot size is ~2 μm). The reflected light was directed into a spectrograph and collected with a CCD camera (Princeton Instruments). The differential reflectance is calculated as $\frac{\Delta R}{R} = \frac{R-R_0}{R_0}$ by using the reflectance spectrum at the highest p-doping region as the reference $R_0$.

### Microwave impedance microscopy measurements
The MIM measurement is performed on a homebuilt cryogenic scanning probe microscope platform. A small microwave excitation of about 0.1 μW at a fixed frequency ~10 GHz is delivered to a chemically etched tungsten tip mounted on a quartz tuning fork. The reflected signal is analyzed to extract the demodulated output channels, MIM-Im and MIM-Re, which are proportional to the imaginary and real parts of the admittance between the tip and sample, respectively. To enhance the MIM signal quality, the tip on the tuning fork is excited to oscillate at a frequency of around 32 kHz with an amplitude of ~8 nm. The resulting oscillation amplitudes of MIM-Im and MIM-Re are then extracted using a lock-in amplifier to yield d(MIM-Im)/dz and d(MIM-Re)/dz, respectively. The d(MIM)/dz signals are free of fluctuating backgrounds, and their behavior is very similar to that of the standard

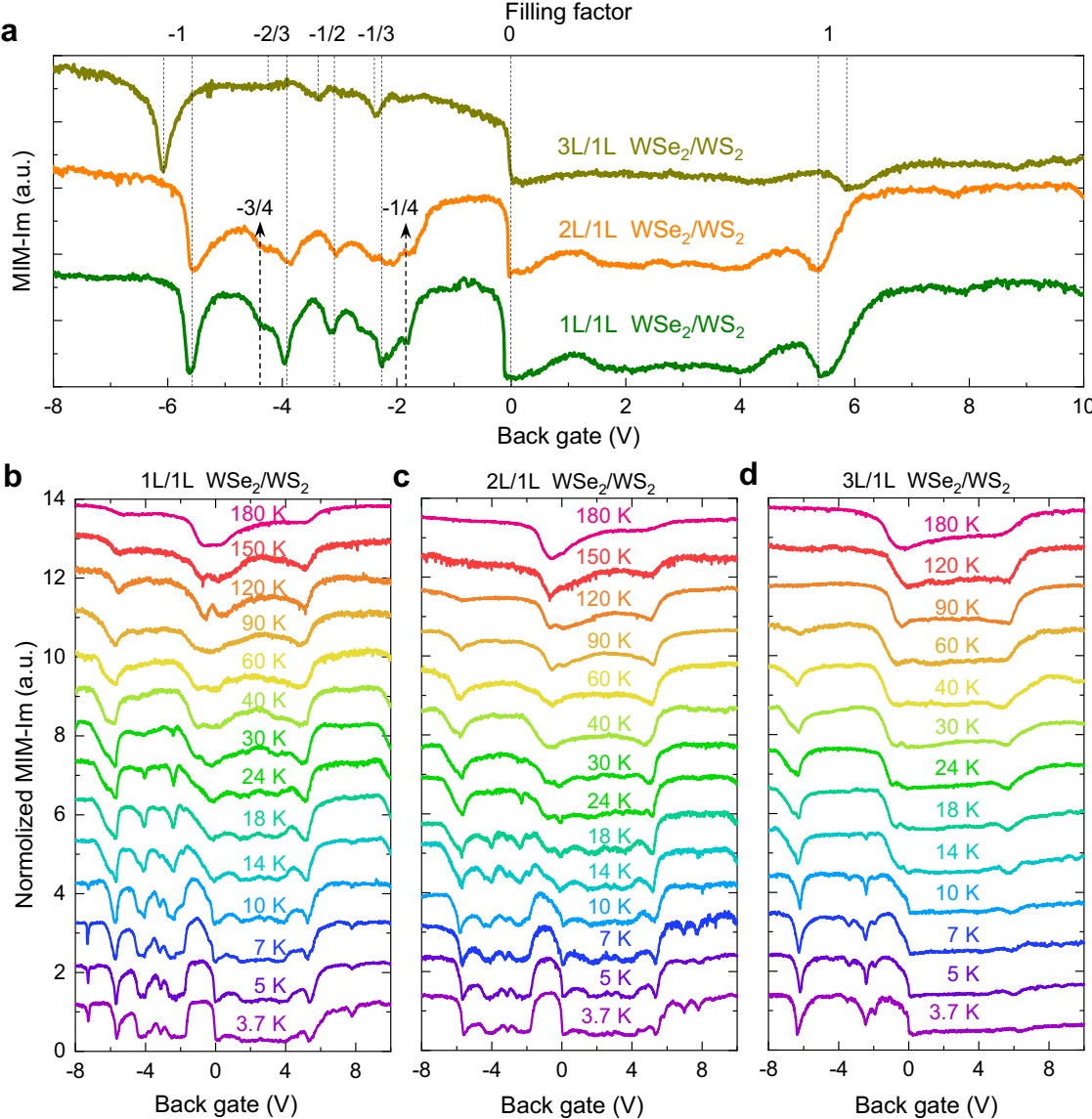

**Fig. 4 | MIM measurements of correlated states in different moiré superlattices.**
**a** MIM spectra as a function of gate voltage for the moiré superlattice of 1 L/1 L
WSe$_2$/WS$_2$ (green), 2 L/1 L WSe$_2$/WS$_2$ (orange), and 3 L/1 L WSe$_2$/WS$_2$ (brown) at
10 K. **b–d** are the temperature-dependent MIM spectra for the moiré superlattice of
1 L/1 L, 2 L/1 L, and 3 L/1 L WSe$_2$/WS$_2$, respectively.

MIM signals. In this paper, we simply refer to d(MIM)/d$z$ as the MIM
signal.

### Estimating the twist angle

Twist angles of the moiré superlattices can be estimated by the carrier
density corresponding to the correlated insulating state at $n = \pm1$, with
a bottom hBN of thickness ~52 nm and dielectric constant 3.5. The 1 L/
1 L WSe$_2$/WS$_2$ and 2 L/1 L WSe$_2$/WS$_2$ regions have a similar moiré peri-
odicity of 7.4 nm and twist angle of 0.9°. For the 3 L/1 L WSe$_2$/WS$_2$
region, the moiré periodicity is 6.0 nm and the twist angle is 1.3°. This
difference is likely caused by a small distortion or wrinkle between
WSe$_2$ and WS$_2$ layers.

### Data availability

Source data are available for this paper. All other data that support the
plots within this paper and other finding of this study are available
from the corresponding author upon reasonable request.

### Code availability

The source code for the numerical simulations is available from the
corresponding author upon reasonable request.

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

## Acknowledgements

We thank professor Feng Wang and professor Chenhao Jin for their helpful discussions. The optical spectroscopy measurements are supported by an AFOSR DURIP award through Grant FA9550-20-1-0179. The device fabrication was supported by the Micro and Nanofabrication Clean Room (MNCR) at Rensselaer Polytechnic Institute (RPI). Z. Lian and S.-F.S. acknowledge support from NYSTAR through Focus Center-NY–RPI Contract C150117. S.-F.S. also acknowledges the support from NSF (Career Grant DMR-1945420 and DMR-2104902) and AFOSR (FA9550-18-1-0312). X.H. and Y.-T.C. acknowledge support from NSF under award DMR- 2104805. Y.S. and C.Z. acknowledge support from NSF PHY-2110212, PHY-1806227, ARO (W911NF17-1-0128), and AFOSR (FA9550-20-1-0220). D.C. acknowledges support from the National Natural Science Foundation of China, Grant number 62004032. S.T. acknowledges support from NSF DMR-1904716, DMR-1838443, CMMI-1933214, and DOE-SC0020653. K.W. and T.T. acknowledge support from the Elemental Strategy Initiative conducted by the MEXT, Japan, Grant Number JPMXP0112101001 and JSPS KAKENHI, Grant Numbers 19H05790 and JP20H00354. L.X. and D.S. acknowledge support from the U.S. Department of Energy (no. DE-FG02-07ER46451) for magneto-spectroscopy measurements performed at the National High Magnetic Field Laboratory, which is supported by the National Science Foundation through NSF/DMR-1644779 and the State of Florida.

## Author contributions

S.-F.S. and Y.-T.C. conceived the project. D.C. and Z.L. fabricated the heterostructure devices and performed the optical spectroscopy measurements. X.H. performed the MIM measurements. M.R. helped with device fabrication. M.B. and S.T. grew the TMDC crystals. T.T. and K.W. grew the BN crystals. C.Z. and Y.S. performed the theoretical calculations. S.-F.S., Y.-T.C., C.Z., Y.S., D.C., Z.L., Z.W., X.H., and L.Y. analyzed the data. S.-F.S. and Y.-T.C. wrote the manuscript with inputs from all authors.

## Competing interests
