## [Peer Review File · Nature Communications]

REVIEWER COMMENTS

Reviewer #1 (Remarks to the Author):

In this work, the authors studied the properties of the moiré excitons and the correlated insulating states in the WSe₂/WS₂ moiré superlattice with additional WSe₂ layers. Based on the optical spectrum of the exciton excitations, they demonstrated the interfacial nature of the moiré couplings. And the influence of additional layers to the correlated insulating states has been addressed by the MIM measurements. The authors propose this multilayer system as a new platform to study the correlation physics.

Although, the experimental results are quite reasonable, the novelty of the manuscript is questionable. Thus, I would not recommend its publication in Nature Communications due to the following comments:

1. The main result of the article is the modulation of the moiré excitons due to intralayer A excitons in the nearby WSe₂ layers. Experimentally, they found nearby A excitons would induce energy shift of the moiré excitons as shown in Figure 1(b) of the main text. As clarified in this article, the moiré excitons are originated from splitting of the A excitons by the moiré coupling. Thus, the energy shift of moiré excitons in 2L and 3L WSe₂ cases could be expected from level repulsion and the interfacial nature of the moiré coupling. Since WSe₂ layers coupled with each other by the van-der Waals force, the energy shift in 3L is comparable with 2L case is obvious.
2. The modulated moiré exciton resonances at certain fillings were attributed to the change of dielectric constant due to the formation of the Mott insulators. Although the excitons are sensitive to the surrounding dielectric environments, the Hubbard bands from fractional filled moiré valance bands would also influence the exciton resonances. It's unclear which one dominates the experimental results.
3. Also, the weakened Mott insulating states were explained by the increased dielectric constant and band widths with additional WSe₂ layers. Since the band widths have been calculated by the effective model. It would be important to estimate how does the Coulomb energy change with additional layers.

Reviewer #2 (Remarks to the Author):

The authors reported an interesting study on Moire excitons in monolayer-few layer TMDC heterostructures. A 1L WS₂ flake is covered by a flake of WSe₂ containing regions of 1L, 2L, and 3L. The two flakes, with near 0 twist angle, form a Moire lattice with an 8-nm period, due to their lattice mismatch. The heterostructure is encapsulated by hBN flakes and is subject to a back gate for doping, through a few-layer graphene flake serving as electrode. The device structure is similar to previously reported ones to study Moire excitons. The key new element is the inclusion of the 2L and 3L regions. Gate dependent low-temperature reflectance spectroscopy and microwave impedance microscopy were performed to probe the electronic and excitonic states of the device. The key findings include the additional exciton peaks in thicker samples, with increased energy differences. Along with theoretical calculations, the authors concluded that the Moire potential is localized at the interface, and the layer thickness can be used to tune the electronic correlation and Moire exciton bands.

I found the topic timely and results interesting. The data reported are of high quality. However, I feel that there are several missing key elements that I wish the authors could consider:

1. It appears that only one device was studied. Although having all three heterostructures on a same sample is definitely an advantage and facilitates their direct comparison, ideally, repeating the study on another independent device appears necessary to solidify the conclusions.

2. There is a lack of sample characterization information. For example, how did the author determine the thickness of the samples? Are there AFM or Raman characterization? Since the authors converted gate voltage to carrier density (Figure 1) and used it in the discussions, do they know the background doping density of the samples?

3. Although reflectance measurement can probe the exciton energies, some complementary photoluminescence experiments would offer a more complete picture of these excitons. Such data are routinely included in most studies on Moire excitons. Hence, if the authors could include such data, it would allow a benchmark comparison with literatures.

4. When calculating carrier density from gate voltage, it appears that the authors didn't consider the different thicknesses of the three regions. Maybe this is valid, but given the broader readership of this journal, it would be helpful if the authors can provide better justification or a reference.

In summary, this is a well designed study and the manuscript is well written. The conclusion is interesting and would indeed introduce a new control knob to the Moire superlattice studies. However, before I can recommend acceptance, I wish the authors could consider improving their manuscript along the lines listed above.

Reviewer #3 (Remarks to the Author):

Dongxue Chen et al. report experimental study of moire excitons and correlated electronic states in multilayer WSe₂ / 1L WS₂ heterostructure. The authors perform combined optical spectroscopy and microwave impedance measurement and show the systematic layer number dependence (1L to 3L) of intralayer exciton spectrum including moire effect and correlated electronic states. As far as I know this is the first systematic study of layer number dependence of moire excitons and correlated electronic states in WSe₂/WS₂ moire system and the experimental effort is impactful for the 2D material research community.

However, I have a major concern about the microscopic mechanism of intralayer exciton and moire exciton hybridization which the authors discuss extensively in the first half of the manuscript.

In line 59 – 64 in Supplementary Note 2, the authors state that moire potential folds the energy bands and compensates the large momentum separation of K – K' valley. However, the momentum difference which moire potential can compensate is only the order of moire reciprocal vector which is much smaller than K – K' momentum difference. I cannot completely deny the possibility, but back-of-the-envelope calculation tells that assuming that the moire lattice constant is ~ 8nm, and intrinsic lattice constant is ~ 0.32nm, order of $N = 8\text{nm}/0.32\text{nm} = 25$ times of perturbation process is required to compensate the momentum difference, which would be unlikely to happen coherently. I tried but failed to come up with different scenario. It is worth to mention that the exchange interaction, which the authors did not discuss explicitly but shown in the last term of the exciton Hamiltonian H_A in Supplementary Note 1, mixes K – K' valley excitons at non-zero momentum. (There is an experimental report regarding the mixing of K – K' valley states of moire exciton, PRX 11, 021027 (2021).)

Another thing I want to mention is that even though interlayer tunneling is forbidden for conduction band electrons due to symmetry as the authors mention, it is allowed for nonzero momentum electrons. Therefore, moire potential would enable the hybridization of intravalley conduction band electrons off resonantly between two layers.

Given that interlayer tunneling is unlikely to hybridize intervalley electron states as I discuss above, it only hybridizes intravalley A exciton and B exciton states with the assist of moire potential off

resonantly. However, A and B excitons do not hybridize within a same layer, so cannot realize A exciton - A exciton hybridization between two layers in this manner.

I agree that the 4x4 matrix model (Eq. #2) captures the experimental observation, but the microscopic physics is unlikely to happen. I recommend the authors to reconsider the microscopic model carefully.

The layer number dependence of correlated electrons is well discussed and the results are convincingly presented.

Minor comments

1. What is the spatial resolution of MIM measurement?
2. For the case of 3L/1L WSe₂/WS₂, $n=\pm 1$ gate voltage condition is different from the other cases (2L/1L or 1L/1L). Is this attributed to the existence of other bands (non-interfacial layer of WSe₂) as shown in Fig. 3c? I did not find the discussion, but this is worth to be mentioned explicitly.
3. Line 90-92: "exciton-exciton interaction" is bit misleading since this paper is discussing exciton coupling but not exciton-exciton interaction which is rather used for many body effect such as nonlinear effect or biexcitonic process. Tunnel coupling of electrons is not regarded as electron - electron interaction for example.
4. Line 286 (Fig.2 caption): "Supplementary Information Note 2" instead of "Supplementary Information Note 1"
5. In Supplementary Note 1, the value of J used in Hamiltonian H_A is missing.
6. In Supplementary Note 1, how the authors quote electron-hole total mass $M=0.75m_e$ from 2D Mater. 2 (2015) 022001? In Fig. 4 of that paper, they define $K^{(1)}_{cb}$ ($m^{(1)}_{cb} \sim 0.28m_e$) and $K^{(1)}_{vb}$ ($m^{(1)}_{cb} \sim 0.36m_e$) as the bright A exciton bands, and the total mass seems to be rather $M \sim 0.64m_e$. However, for the dark A exciton bands, $K^{(2)}_{cb}$ ($m^{(2)}_{cb} \sim 0.39m_e$) and $K^{(1)}_{vb}$ ($m^{(1)}_{cb} \sim 0.36m_e$), the total mass seems to be $M \sim 0.75m_e$, which agrees with what the authors quote. Probably the theoretical prediction and experimental measurement of the effective mass would not be precise enough to distinguish this difference, but it is nice to be quoted properly.
7. Line 88 in Supplementary Note 2: Why the authors chose $t_{III} = -0.04i$ as an imaginary number? Whether $t_{III} = -0.04$ or $-0.04i$ does not make difference in terms of the eigen values of the matrix (2).
8. Line 51 in Supplementary Note 2: "shown in Fig. S1" instead of "shown in Fig. S2"
9. Line 106-107 in Supplementary Note 3: V_0 and ϕ , which are related to moire potential, are not quoted from early literatures [6, 7] (Nat. Commun. 4, 15 (2013) and PRL 108, 196802 (2012)). I recommend the authors to add references for these numbers also.

We sincerely thank the reviewers for their time and efforts. We also greatly appreciate the reviewers' recognition of our work and their constructive feedback. Here we provide a point-to-point reply to their comments in the following (blue colored). We have also revised our manuscript accordingly, with the revision highlighted in the main text and SI for the reviewers' convenience.

Here we list the major revisions we have made in the main text and SI.

1. We have developed the microscopic mechanism of exciton hybridization and added the corresponding discussion in the main text and Supplementary Note 2.
2. We have included an extensive discussion of layer characterization of WSe₂ in Supplementary Note 4.
3. We have included extensive data from other devices in Supplementary Note 5.

With these revisions, we believe that our manuscript is ready for publication in Nature Communications. We thank all three reviewers for helping us improve our manuscript.

REVIEWER COMMENTS

Reviewer #1 (Remarks to the Author):

In this work, the authors studied the properties of the moiré excitons and the correlated insulating states in the WSe₂/WS₂ moiré superlattice with additional WSe₂ layers. Based on the optical spectrum of the exciton excitations, they demonstrated the interfacial nature of the moiré couplings. And the influence of additional layers to the correlated insulating states has been addressed by the MIM measurements. The authors propose this multilayer system as a new platform to study the correlation physics.

Although, the experimental results are quite reasonable, the novelty of the manuscript is questionable. Thus, I would not recommend its publication in Nature Communications due to the following comments:

Response: We thank the reviewer for the efforts in reviewing our manuscript. Here, we take this opportunity to elaborate on the novelty of our work. We have also revised our manuscript accordingly to emphasize some of the discussions below. We thank the reviewer for helping us improve our manuscript.

To the best of our knowledge, our work is the first study of TMDC heterojunction moiré superlattices with a layer degree of freedom.

In this work, we find that the added layer(s) significantly modulate moiré excitons' resonance energies. We have also developed the microscopic mechanism of the moiré exciton modulation: hybridization of moiré excitons and intralayer excitons from the added layer(s). Both experimental

tuning and theoretical understanding of the moiré exciton are critical for future investigation of excitonic physics in TMDC moiré superlattices.

We also find that the added layer(s) reduce the correlation strength in the moiré superlattices. We have theoretically calculated the bandwidth of the moiré miniband, which is consistent with the experimental observations. We emphasize that, even the reduced correlation strength in 3L/1L WSe₂/WS₂ moiré superlattice is still much stronger than that in graphene moiré systems, evidenced by the Mott transition temperature. Considering the valley-spin-layer locking, the multilayer TMDC moiré superlattices provide an exciting platform to engineer new correlated physics, especially considering the valley contrasting excitons.

Finally, our study reveals the highly interfacial nature of the moiré interaction, which provides critical guidance for future studies and applications of moiré physics.

In summary, we believe that our results provide stimulating results for researchers working on excitons, moiré superlattices, and correlated physics in 2D, and the multilayer TMDC moiré superlattices will provide a new exciting platform for excitonic physics and correlated states.

Below we provide a point-to-point reply to the reviewer's comments:

1. The main result of the article is the modulation of the moiré excitons due to intralayer A excitons in the nearby WSe₂ layers. Experimentally, they found nearby A excitons would induce energy shift of the moiré excitons as shown in Figure 1(b) of the main text. As clarified in this article, the moiré excitons are originated from splitting of the A excitons by the moiré coupling. Thus, the energy shift of moiré excitons in 2L and 3L WSe₂ cases could be expected from level repulsion and the interfacial nature of the moiré coupling. Since WSe₂ layers coupled with each other by the van-der Waals force, the energy shift in 3L is comparable with 2L case is obvious.

Response: As we mentioned above, the modulation of the moiré excitons is one of our major results.

The large moiré exciton modulation, especially when the WSe₂ is changed from 1L to 2L, is significant. Such large tunability of the moiré excitons will be important for applications, and the understanding will also provide critical insight on moiré excitons. We have developed a hybridization mechanism for the moiré exciton with the intralayer A exciton, which well describes our observation. To the best of our knowledge, this is the first observation of such hybridization. It can also be seen from our reply to reviewer 3's comments that the mechanism is nontrivial and highly intriguing.

In addition, the optical spectra of the new exciton resonance close to the intralayer exciton A exciton is a direct consequence of the interfacial nature of moiré coupling, and we believe our work is the first report of that. The interfacial nature of the moiré coupling is critical for designing future systems to study and utilize moiré excitons.

Finally, the exact nature of the moiré exciton in WS_2/WSe_2 superlattice is of great interest and under active investigation. Our experimental observations of the moiré exciton modulation could stimulate and help further theoretical efforts in understanding the moiré excitons.

2. The modulated moiré exciton resonances at certain fillings were attributed to the change of dielectric constant due to the formation of the Mott insulators. Although the excitons are sensitive to the surrounding dielectric environments, the Hubbard bands from fractional filled moiré valence bands would also influence the exciton resonances. It's unclear which one dominates the experimental results.

We wonder if the reviewer is referring to Hubbard bands at the half-filled valence bands. If so, we agree with the reviewer that the gap opening due to Hubbard band formation can potentially affect the moiré excitons as well, which is stated in the main text “The modulation of the moiré excitons at the $n=\pm 1$ is likely due to the dielectric constant change and gap opening associated with the Mott insulator states.”

The exact mechanism of how the formation of correlated states affects the moiré exciton is intriguing but it is not the focus of our manuscript. In fact, the mechanism remains unclear, despite that it has been used to investigate the correlated states in quite a few recent works [2, 3]. Although the exact mechanism is out of the scope of this manuscript, our work provides a clear layer dependence of the electron correlation through temperature-dependent MIM measurements. The corresponding moiré exciton spectra shown in this work, therefore, will be helpful for further investigation of the mechanism.

3. Also, the weakened Mott insulating states were explained by the increased dielectric constant and band widths with additional WSe_2 layers. Since the band widths have been calculated by the effective model. It would be important to estimate how does the Coulomb energy change with additional layers.

Estimating the Coulomb interaction and how it affects the correlated states is challenging, as the effective dielectric constant of the structure is difficult to calculate, considering the 2D nature of the added layers. But that is exactly the beauty of our work, as we experimentally provide high quality data that clearly show the layer dependence. Our results, therefore, provide critical information for this new platform and will stimulate future theoretical exploration.

We also emphasize that, although one might expect the reduced Mott transition temperature with added layers, it is not clear how much reduction there will be. Even more critically, it is not clear whether the correlated states will survive with the added layers. Our work is the first to show that the correlated states persist with the added layers. Even in the $3\text{L}/1\text{L } \text{WSe}_2/\text{WS}_2$ region, where the correlation strength is much reduced compared with the $1\text{L}/1\text{L } \text{WSe}_2/\text{WS}_2$, the electron correlation is still much more robust than that of graphene moiré superlattices, with the Mott transition

temperature of ~60 K compared with the ~4 K [4] in the graphene moiré superlattices. This is exciting because one can envision controlling correlated electrons' distribution in different layers to engineer new quantum states, and it is not feasible in the 1L/1L WSe₂/WS₂, in which the large band offset between WSe₂/WS₂ strictly restrains electrons to reside in the WS₂ layer and holes in the WSe₂ layer. We have revised manuscript accordingly to reflect this discussion.

Reviewer #2 (Remarks to the Author):

The authors reported an interesting study on Moiré excitons in monolayer-few layer TMDC heterostructures. A 1L WS₂ flake is covered by a flake of WSe₂ containing regions of 1L, 2L, and 3L. The two flakes, with near 0 twist angle, form a Moiré lattice with an 8-nm period, due to their lattice mismatch. The heterostructure is encapsulated by hBN flakes and is subject to a back gate for doping, through a few-layer graphene flake serving as electrode. The device structure is similar to previously reported ones to study Moiré excitons. The key new element is the inclusion of the 2L and 3L regions. Gate dependent low-temperature reflectance spectroscopy and microwave impedance microscopy were performed to probe the electronic and excitonic states of the device. The key findings include the additional exciton peaks in thicker samples, with increased energy differences. Along with theoretical calculations, the authors concluded that the Moiré potential is localized at the interface, and the layer thickness can be used to tune the electronic correlation and Moiré exciton bands.

I found the topic timely and results interesting. The data reported are of high quality. However, I feel that there are several missing key elements that I wish the authors could consider:

Response: We greatly appreciate the reviewer's recognition of our work, and we provide a point-to-point reply to all the questions by the reviewer in the following.

1. It appears that only one device was studied. Although having all three heterostructures on a same sample is definitely an advantage and facilitates their direct comparison, ideally, repeating the study on another independent device appears necessary to solidify the conclusions.

Response: Although we only presented the data from one device in the main text, the observations are robust. We have studied the moiré excitons behavior of 9 different samples (not necessary all include 1L/1L, 2L/1L, and 3L/1L WSe₂/WS₂ regions as it is more difficult to fabricate), and all the data are consistent with the one presented in the main text. We have summarized the results in Table R1 and included it in the revised Supplementary Information (Supplementary Table 1). We also include it for the reviewer's convenience.

Table R1. Exciton resonances from different samples.

Device	Region	X_I (eV)	X_{II} (eV)	X_{III} (eV)	X_{IV} (eV)	Device	Region	X_I (eV)	X_{II} (eV)	X_{III} (eV)	X_{IV} (eV)
D1	1L/1L WSe ₂ /WS ₂	1.662	1.715	1.753		D5	2L/1L WSe ₂ /WS ₂	1.657	1.742	1.794	1.705
	2L/1L WSe ₂ /WS ₂	1.642	1.728	1.793	1.693	D6	1L/1L WSe ₂ /WS ₂	1.687	1.737	1.772	
	3L/1L WSe ₂ /WS ₂	1.645	1.730	1.785	1.677		2L/1L WSe ₂ /WS ₂	1.665	1.757	1.797	1.701
D2	1L/1L WSe ₂ /WS ₂	1.652	1.699	1.739		D7	1L/1L WS ₂ /WSe ₂	1.687	1.725	1.765	
	2L/1L WSe ₂ /WS ₂	1.657	1.737	1.797	1.703		1L/2L WS ₂ /WSe ₂	1.674	1.747	1.799	1.710
	3L/1L WSe ₂ /WS ₂	1.642	1.728	1.789	1.678	D8	1L/1L WSe ₂ /WS ₂	1.685	1.723	1.763	
D3	2L/1L WSe ₂ /WS ₂	1.683	1.772	1.811	1.712		2L/1L WSe ₂ /WS ₂	1.666	1.753	1.795	1.706
	3L/1L WSe ₂ /WS ₂	1.669	1.756	1.824	1.717	D9	1L/1L WSe ₂ /WS ₂	1.687	1.748	1.792	
D4	2L/1L WSe ₂ /WS ₂	1.661	1.755	1.801	1.697		2L/1L WSe ₂ /WS ₂	1.652	1.754	1.803	1.704

In addition, we have added the optical spectra of device D2 in the revised Supplementary Information. Device D2 is similar to the device D1 (data presented in the main text) and has well-defined 1L/1L, 2L/1L and 3L/1L WSe₂/WS₂ regions, with the optical image shown in Supplementary Figure 3b. The optical spectra from D2 are similar to what we present in the main text, which we include in the revised Supplementary Figure 7. For the reviewer's convenience, we also include the data here as Fig. R1, which show consistent results compared with the device shown in the main text (D1, Fig. 1).

Figure R1. (a), (b) and (c) are the differential reflectance spectra as a function of the gate voltage in the regions of 1L/1L, 2L/1L, and 3L/1L WSe₂/WS₂, respectively. All data were taken at 10 K. (d), (e) and (f) are differential reflectance spectra for different regions at zero gate voltage. This figure is also added to Supplementary Information as Supplementary Figure 8.

2. There is a lack of sample characterization information. For example, how did the author determine the thickness of the samples? Are there AFM or Raman characterization? Since the authors converted gate voltage to carrier density (Figure 1) and used it in the discussions, do they know the background doping density of the samples?

Response: Over the past few years, we have developed a systematic method to determine the layer number of WSe₂ in the few-layer regime. The optical reflectance from a thin flake of 2D materials is related to its absorption [5,6] and thus a powerful method to determine the thickness of the 2D materials such as WSe₂. The microscope image of three WSe₂ flakes (including a standard WSe₂ flake, device D1, and device D2) are shown in Fig. R2. We can apply optical contrast analysis method [5,6] to identify the thickness by numerically analyzing the RGB optical images of the samples.

The microscope images of the standard WSe₂ flake, the WSe₂ flake used in D1, and D2 are shown in Figure R4a-c. The reflectance contrast (R contrast, defined as $(R_{\text{substrate}} - R_{\text{sample}})/R_{\text{substrate}}$) for these three pieces of WSe₂ flake is also calculated (the red channel), and the values for different layer regions are plotted in Fig. R2d. It is evident that WSe₂ regions with a certain layer number share similar R contrast, while the increase of one single layer will increase the R contrast by one “step”. Thus, by comparing with the R contrasts of standard WSe₂ flake and the WSe₂ flakes used in D1 and D2, we can accurately confirm the number of layers of WSe₂.

Figure R2. Reflectance contrast of (a) standard WSe₂, (b) WSe₂ prepared for D1 and (c) WSe₂ prepared for D2. (d) Reflectance contrast of the standard WSe₂ sample, the WSe₂ flake for D1 and D2. This figure is also added to Supplementary Information as Supplementary Figure 5.

We further confirm the layer thickness through AFM measurements (Fig. R3). For the standard WSe₂ sample, we overlay the AFM topography measurements with the microscope image. It is evident that each layer adds to about 0.74 ± 0.03 nm in height, consistent with previous reports [7].

Figure R3. Optical images of (a) standard WSe₂, (b) WSe₂ prepared for D1 and (c) WSe₂ prepared for D2. (d) AFM measurement results of the standard WSe₂ flake. This figure is also added to Supplementary Information as Supplementary Figure 6.

Finally, the layer assignment is consistent with the Raman measurements. Fig. R4 shows the Raman spectra taken from the different regions on the standard sample using 532 nm laser excitation. A Raman peak is observed at 308.0 cm⁻¹ on the 2L WSe₂ region and at 307.7 cm⁻¹ on the 3L WSe₂ region, yet it is absent on the monolayer WSe₂ region, as shown in Fig. R4c. We also observe a splitting between the A_{1g} mode and the E_{2g}¹ mode on the 2L WSe₂ region and the 3L WSe₂ region, shown in Fig. R4a. All these observations are consistent with the Raman spectra reported in Ref. [8].

Figure R4. Raman spectra of 1L, 2L and 3L WSe₂ from the standard sample. This Figure is added as Supplementary Figure 7 in the revised Supplementary Information.

The initial doping of the samples is close to zero, which can be shown by the device behavior at zero gate voltage (MIM data in Fig. 4 and optical reflection spectra in Fig. 1 of the main text).

3. Although reflectance measurement can probe the exciton energies, some complementary photoluminescence experiments would offer a more complete picture of these excitons. Such data are routinely included in most studies on Moiré excitons. Hence, if the authors could include such data, it would allow a benchmark comparison with literatures.

Response: We thank the reviewer for the suggestion. Here we include the gate-dependent PL spectra of the interlayer excitons for the device shown in the main text (device D1) in Fig. R5. The data from the 1L/1L WSe₂/WS₂ clearly show the emergence of the correlated insulating states at fractional fillings (see Fig. R5a). The n=1 (-1) is the Mott insulator state for the half-filling corresponding to one electron (hole) per moiré superlattice. The fillings at 1/3, 2/3, 1/4, and 1/2 correspond to correlated insulating states we reported earlier [9,10], and the capability of revealing them directly in the PL spectra demonstrates the high quality of the data.

The PL of interlayer excitons is weaker in 2L/1L and 3L/1L WSe₂/WS₂ regions, as shown in Fig. R5b, c. The exact mechanism is still under investigation. Although PL spectra are of high sensitivity, it is also incoherent in nature, and the mechanism could be complicated. Unlike reflectance spectroscopy, which is directly linked to absorption, both absorption and emission processes are important for PL emission. For the 1L/1L WSe₂/WS₂ superlattice system, since moiré modulation of the intralayer excitons is robust and reasonably well understood [11], we focus on using reflectance spectroscopy to understand the modulation due to layer numbers in the main text.

We also include the PL spectra in Fig. R5 in the revised Supplementary Information as Supplementary Figure 9.

Figure R5. Gate-dependent PL spectra from 1L/1L, 2L/1L, and 3L/1L WSe₂/WS₂ region of device D1. All data were taken at 4.5 K.

4. When calculating carrier density from gate voltage, it appears that the authors didn't consider the different thicknesses of the three regions. Maybe this is valid, but given the broader readership of this journal, it would be helpful if the authors can provide better justification or a reference.

Response: We thank the reviewer for the suggestion. The AFM measurements of device D1 show a uniform thickness of 52 nm, much larger than the thickness difference between 1L/2L/3L (one layer is about 0.74 nm, see reply to comment 2). For a geometry capacitance model, $n = \epsilon V / ed$, where n is the density of carrier, ϵ is the dielectric constant, V is the voltage effectively applied to the sample, e is the elementary charge of the electron, and d is the thickness of the dielectric. In this case, d is approximately 52 nm, with the difference between different layers having negligible effects.

In summary, this is a well designed study and the manuscript is well written. The conclusion is interesting and would indeed introduce a new control knob to the Moiré superlattice studies. However, before I can recommend acceptance, I wish the authors could consider improving their manuscript along the lines listed above.

Response: Again, we greatly appreciate the reviewer's recognition of our work. We also thank the reviewer for the comments and suggestions that help us to improve our manuscript.

Reviewer #3 (Remarks to the Author):

Dongxue Chen et al. report experimental study of Moiré excitons and correlated electronic states in multilayer WSe₂ / 1L WS₂ heterostructure. The authors perform combined optical spectroscopy and microwave impedance measurement and show the systematic layer number dependence (1L to 3L) of intralayer exciton spectrum including Moiré effect and correlated electronic states. As far as I know this is the first systematic study of layer number dependence of Moiré excitons and correlated electronic states in WSe₂/WS₂ Moiré system and the experimental effort is impactful for the 2D material research community.

Response: We thank the reviewer for the positive evaluation of our work.

However, I have a major concern about the microscopic mechanism of intralayer exciton and Moiré exciton hybridization which the authors discuss extensively in the first half of the manuscript.

In line 59 – 64 in Supplementary Note 2, the authors state that Moiré potential folds the energy bands and compensates the large momentum separation of K – K' valley. However, the momentum difference which Moiré potential can compensate is only the order of Moiré reciprocal vector which is much smaller than K – K' momentum difference. I cannot completely deny the possibility, but back-of-the-envelope calculation tells that assuming that the Moiré lattice constant is ~ 8nm, and intrinsic lattice constant is ~ 0.32nm, order of $N = 8\text{nm}/0.32\text{nm} = 25$ times of perturbation process is required to compensate the momentum difference, which would be unlikely to happen coherently. I tried but failed to come up with different scenario. It is worth to mention that the exchange interaction, which the authors did not discuss explicitly but shown in the last term of the exciton Hamiltonian H_A in Supplementary Note 1, mixes K – K' valley excitons at non-zero momentum. (There is an experimental report regarding the mixing of K – K' valley states of Moiré exciton, PRX 11, 021027 (2021).)

Response: We thank the reviewer for the critical comment, which helps us to reconsider and figure out the microscopic mechanism of interlayer hybridization between moiré and intralayer A excitons. We find that the hybridization is enabled by the combination of moiré-potential-induced Umklapp scattering and intervalley exchange interaction. To show this explicitly, let us first examine the nature of moiré excitons. In 1L/1L WSe₂/WS₂, the moiré potential can hybridize two exciton states if their momenta are differed by the moiré reciprocal lattice vectors G_i , as displayed in Fig. R1a. Here we show that the WSe₂ bright intralayer A exciton with zero center-of-mass momentum couples to the six Umklapp exciton states in the first shell. These Umklapp states are originally momentum dark, and the intervalley exchange interaction hybridizes the K and K' valley excitons at nonzero momentum that splits the exciton dispersion into two branches [12-14], as mentioned by the reviewer. In Fig. R1b, we show the exciton minibands at the moiré potential $V = 0$ meV such that the bare WSe₂ intralayer A exciton bands (which are represented by the red dashed curves) are trivially folded into the mini Brillouin zone (MBZ). The bright A exciton X_A

and the six degenerate Umklapp states from the lower branch of the bare dispersion are coincident at the γ point of MBZ and are highlighted in Fig. R1b. Now we turn on a weak $V = 5$ meV and the hybridization between X_A and Umklapp states splits the six-fold degenerate point into two double degenerate points and two nondegenerate points, as shown in Fig. R1c. As the moiré potential increases further to $V = 25$ meV, the well-separated lowest three double degenerate points give rise to the moiré excitons X_I , X_{II} , and X_{III} , which are highlighted in Fig. R1d. The moiré excitons inherit the component of X_A through the Umklapp scattering and are bright. To show this, we plot the valley pseudospin vectors of the moiré exciton states X_I , X_{II} , and X_{III} in Fig. R1e-g, respectively. The north and south poles of the Bloch sphere are the valley polarized bright A exciton states $|X_{A,K}\rangle$ and $|X_{A,K'}\rangle$. Each moiré exciton is formed by two degenerate Kramers pairs with opposite valley pseudospins. Apparently, the pseudospin vectors have finite components along the direction of $|X_{A,K}\rangle$ or $|X_{A,K'}\rangle$. On the other hand, the exciton states from the two nondegenerate points are orthogonal to X_A and therefore dark. Because the hybridization between X_A and Umklapp states from the upper branch of the bare dispersion is suppressed by the larger energy gap, the higher-energy exciton states at γ are also dark.

Now we have established that the moiré exciton is a mixture of the K and K' valley excitons. On the other hand, in 2L/1L and 3L/1L WSe₂/WS₂, the moiré coupling is highly localized at the WSe₂/WS₂ interface, and the valley remains approximately a good quantum number of the bright intralayer excitons in the added WSe₂ layers away from the interface. As a result, the moiré excitons at the interface can always hybridize with the valley excitons in added WSe₂ layers, whether they are in K or K' valley. The interlayer hybridization decreases exponentially with the interlayer distance.

Figure R6. (a) Schematic Umklapp scattering induced by the moiré potential. The black hexagons are the MBZs. (b-d) Exciton minibands of 1L/1L WSe₂/WS₂ with $V = 0, 5, 25$ meV. The red dashed curves represent the bare dispersion of WSe₂ intralayer A exciton. The energy bands are plotted along the green dashed path in (a). (e-g) Pseudospin vectors of the moiré exciton states X_I ,

X_{II} , and X_{III} in (d), respectively. The north and south poles of the Bloch sphere are the valley polarized bright A exciton states $|X_{A,K}\rangle$ and $|X_{A,K'}\rangle$.

Another thing I want to mention is that even though interlayer tunneling is forbidden for conduction band electrons due to symmetry as the authors mention, it is allowed for nonzero momentum electrons. Therefore, Moiré potential would enable the hybridization of intravalley conduction band electrons off resonantly between two layers. Given that interlayer tunneling is unlikely to hybridize intervalley electron states as I discuss above, it only hybridizes intravalley A exciton and B exciton states with the assist of Moiré potential off resonantly. However, A and B excitons do not hybridize within a same layer, so cannot realize A exciton - A exciton hybridization between two layers in this manner.

Response: We agree with the reviewer that A and B excitons do not hybridize within the same layer. Therefore, the A exciton – A exciton hybridization we observed cannot be mediated by the A exciton – B exciton hybridization, which is not considered in our analysis.

I agree that the 4x4 matrix model (Eq. #(2)) captures the experimental observation, but the microscopic physics is unlikely to happen. I recommend the authors to reconsider the microscopic model carefully.

Response: We thank the reviewer for the suggestion. We hope our detailed explanation above can convince the reviewer that the moiré-potential-induced Umklapp scattering and intervalley exchange interaction can hybridize the moiré exciton and intralayer A exciton which can be captured by the 4×4 model.

The layer number dependence of correlated electrons is well discussed and the results are convincingly presented.

Response: We greatly appreciate the reviewer’s positive comments on our work, and we particularly thank the reviewer for the constructive advice that help us to improve our manuscript.

Minor comments

1. What is the spatial resolution of MIM measurement?

Response: The spatial resolution of MIM measurement is about 100 nm, limited by the AFM tip size.

2. For the case of 3L/1L WSe₂/WS₂, $n=\pm 1$ gate voltage condition is different from the other cases (2L/1L or 1L/1L). Is this attributed to the existence of other bands (non-interfacial layer of WSe₂) as shown in Fig. 3c? I did not find the discussion, but this is worth to be mentioned explicitly.

Response: We are not exactly sure about the mechanism causing the difference in the gate voltage. But since it is symmetric for $n=+1$ and $n=-1$, the most likely scenario is a small difference in the twist angle in 3L/1L region compared to other regions. From the gate voltage values of the $n=+1$ or $n=-1$ states, we have estimated the twist angles, which is $\sim 1.3^\circ$ for the 3L/1L region, and $\sim 0.9^\circ$ for the 1L/1L and 2L/1L regions. The small difference can be due to a small distortion or wrinkle between the 3L/1L and other regions. It is unlikely due to other bands since the doped electrons reside in the WS₂ layer while the holes reside in the WSe₂ layer, which would unlikely to be symmetric as there is only one layer of WS₂. We will explore it in our future study. We have added a brief discussion in the revised manuscript.

3. Line 90-92: “exciton-exciton interaction” is bit misleading since this paper is discussing exciton coupling but not exciton-exciton interaction which is rather used for many body effect such as nonlinear effect or biexcitonic process. Tunnel coupling of electrons is not regarded as electron – electron interaction for example.

Response: we agree with the reviewer on this suggestion. We have revised manuscript accordingly to avoid possible confusion.

4. Line 286 (Fig.2 caption): “Supplementary Information Note 2” instead of “Supplementary Information Note 1”

Response: We thank the reviewer for spotting the mistake. We have corrected it in the revised manuscript.

5. In Supplementary Note 1, the value of J used in Hamiltonian H_A is missing.

Response: We thank the reviewer for pointing this out. $J = 0.04 \text{ eV}\cdot\text{nm}$ is specified in the revised version and is referred to that used in *Nature* 567, 76–80 (2019), which is cited as Ref. [6] in the Supplementary Information.

6. In Supplementary Note 1, how the authors quote electron-hole total mass $M=0.75m_e$ from 2D Mater. 2 (2015) 022001? In Fig. 4 of that paper, they define $K^{(1)}_{cb}$ ($m^{(1)}_{cb} \sim 0.28m_e$) and $K^{(1)}_{vb}$ ($m^{(1)}_{vb} \sim 0.36m_e$) as the bright A exciton bands, and the total mass seems to be rather $M \sim 0.64m_e$. However, for the dark A exciton bands, $K^{(2)}_{cb}$ ($m^{(2)}_{cb} \sim 0.39m_e$) and $K^{(1)}_{vb}$ ($m^{(1)}_{vb} \sim 0.36m_e$), the total mass seems to be $M \sim 0.75m_e$, which agrees with what the authors quote. Probably the theoretical prediction and experimental measurement of the effective mass would not be precise enough to distinguish this difference, but it is nice to be quoted properly.

Response: We agree with the reviewer that the electron-hole total mass should be $M = 0.64m_e$. The total mass has been corrected and the exciton spectra in Fig. 2 have been recalculated for $M = 0.64m_e$ in the revised version. We thank the reviewer for the suggestion.

7. Line 88 in Supplementary Note 2: Why the authors chose $t_{\text{III}} = -0.04i$ as an imaginary number? Whether $t_{\text{III}} = -0.04$ or $-0.04i$ does not make difference in terms of the eigen values of the matrix (2).

Response: We thank the reviewer for the question. t_{μ} (where $\mu = \text{I, II, III}$) denotes the coupling between different exciton states and its value is determined by the overlap integral of exciton wavefunctions. In principle, t_{μ} is a complex number and its phase depends on the phase difference between exciton wavefunctions. In the phenomenological model Hamiltonian Eq. #(2), we chose these parameters to match the optical absorption spectra observed in the experiment and we found that $t_{\text{III}} = -0.04i$ can better reproduce the experimental results. Although $t_{\text{III}} = -0.04$ does not change the eigenvalues of the Hamiltonian Eq. #(2), the eigenstates are changed that affects the resonant peak height, as shown in Fig. R7. Here the black dashed lines are the absorption spectra for $t_{\text{III}} = -0.04$. In contrast to those for $t_{\text{III}} = -0.04i$, the resonant peaks of $X_{\text{I}}^{2\text{L}/3\text{L}}$, $X_{\text{II}}^{2\text{L}/3\text{L}}$, and $X_{\text{III}}^{2\text{L}/3\text{L}}$ have similar height in both 2L/1L and 3L/1L WSe_2/WS_2 . However, the experimental results in Fig. 1 of the main text show that the resonant peak of $X_{\text{I}}^{2\text{L}/3\text{L}}$ is always dominant over that of $X_{\text{II}}^{2\text{L}/3\text{L}}$ and $X_{\text{III}}^{2\text{L}/3\text{L}}$. Therefore, we chose $t_{\text{III}} = -0.04i$. To explicitly calculate the coupling constant t_{μ} , one needs to know the exact exciton wavefunctions and their distribution in the out-of-plane direction, which is beyond the scope of this phenomenological approach.

Figure R7. Optical absorption spectra in 1L/1L, 2L/1L, and 3L/1L WSe₂/WS₂. The solid lines are obtained for $t_{\text{III}} = -0.04i$, while the dashed lines are obtained for $t_{\text{III}} = -0.04$.

8. Line 51 in Supplementary Note 2: “shown in Fig. S1” instead of “shown in Fig. S2”

We thank the reviewer for catching the typo and we have corrected it in the revised manuscript. We have also gone through the manuscript carefully to correct other typos and grammar mistakes. We thank the reviewer for reading our manuscript carefully and his/her constructive comments that helped to improve our manuscript significantly.

9. Line 106-107 in Supplementary Note 3: V_0 and ϕ , which are related to Moiré potential, are not quoted from early literatures [6, 7] (Nat. Commun. 4, 15 (2013) and PRL 108, 196802 (2012)). I recommend the authors to add references for these numbers also.

Response: We thank the reviewer for the suggestion. We adopt $V_0 = 15$ meV and $\phi = \pi/4$ from *PRB 102, 201115 (2020)*, which is cited as Ref. [10] in the revised version.

References

1. Mit H. Naik *et al.* Nature of novel moiré exciton states in WSe₂/WS₂ heterobilayers. arXiv:2201.02562 (2022).
2. Emma C. Regan *et al.* Mott and generalized Wigner crystal states in WSe₂/WS₂ moiré superlattices. *Nature* 359-363 (2020).
3. Yanhao Tang *et al.* Simulation of Hubbard model physics in WSe₂/WS₂ moiré superlattices. *Nature* 353–358 (2020).
4. Yuan Cao *et al.* Correlated insulator behaviour at half-filling in magic-angle graphene superlattices. *Nature* 556, 80–84 (2018).
5. Dan Bing *et al.* Optical contrast for identifying the thickness of two-dimensional materials. *Optics Communications* 406, 128-138 (2018).
6. Najme S. Taghavi *et al.* Thickness determination of MoS₂, MoSe₂, WS₂ and WSe₂ on transparent stamps used for deterministic transfer of 2D materials. *Nano Research* 12, 1691–1695 (2019).
7. Ali Han *et al.* Growth of 2H stacked WSe₂ bilayers on sapphire. *Nanoscale Horiz* 4, 1434-1442 (2019).

8. Weijie Zhao *et al.* Lattice dynamics in mono- and few-layer sheets of WS₂ and WSe₂. *Nanoscale* **5**, 9677-9683 (2013).
9. Xiong Huang *et al.* Correlated insulating states at fractional fillings of the WS₂/WSe₂ moiré lattice. *Nature Physics* **17**, 715–719 (2021).
10. Shengnan Miao *et al.* Strong interaction between interlayer excitons and correlated electrons in WSe₂/WS₂ moiré superlattice. *Nature Communications* **12**, 3608 (2021).
11. Chenhao Jin *et al.* Observation of moiré excitons in WSe₂/WS₂ heterostructure superlattices. *Nature* **567**, 76–80 (2019).
12. Yuya Shimazaki *et al.* Optical signatures of periodic charge distribution in a Mott-like correlated insulator state. *Physics Review X* **11**, 021027 (2021).
13. Hongyi Yu *et al.* Dirac cones and Dirac saddle points of bright excitons in monolayer transition metal dichalcogenides. *Nature Communications* **5**, 3876 (2014).
14. Fengcheng Wu *et al.* Exciton band structure of monolayer MoS₂. *Physics Review B* **91**, 075310 (2015).

REVIEWER COMMENTS

Reviewer #2 (Remarks to the Author):

The authors have provided additional data and analysis in response to my technical comments. In my view, these responses and revisions are adequate. I support acceptance of the manuscript.

Reviewer #3 (Remarks to the Author):

I appreciate the authors for their detailed reply.

I am confused with the double headed arrows in Bloch sphere in Supplementary Figure 2 e-f which are not well defined (usually pseudospin is represented as single headed arrow). Let me interpret these as pairs of eigenstates, which are energetically degenerate (otherwise there should be time reversal symmetry breaking due to the finite $|X_{A,K}\rangle$ population which has non-zero angular momentum). Since all pseudospin states in the Bloch sphere can be represented with the superposition of these paired eigenstates which are energetically degenerate, that means there is essentially no valley mixing (all pseudospin states are energetically degenerate). I imagine that the authors solved the problem numerically and obtained these pairs of eigenstates with random orientation in Supplementary Figure 2e-f. I suspect that the numerical approach which the authors took hinders the essential understanding of the problem. I rather recommend to solve the problem analytically with some approximation to clarify the symmetry of the problem (For example, what determines the axis of the pseudospin states in Supplementary Fig. 2e-f?)

I trust the experimental results with considerable efforts and the paper is worth to be published. However, I am afraid that this unclear discussion which would mislead the community in future. Compared to the experimental efforts which the authors made, solving this problem analytically is not a big deal and I hope that the authors convince the readers properly with bit more efforts.

We sincerely thank the reviewers for their time and efforts. Following the suggestion of Reviewer #3, we have performed analytical calculations to reveal the nature of moiré excitons which are detailed in our reply below. An extensive discussion about the possible mechanism for interlayer hybridization between different excitonic states are also provided in the reply. Our observation is well explained by a phenomenological model using the 4 by 4 matrix, while the proposed exciton hybridization provides one possible microscopic mechanism of the phenomenological model. We have also revised our manuscript accordingly. With these revisions, we believe that we have addressed the reviewer's question and our manuscript is now ready for the publication in Nature Communications.

REVIEWER COMMENTS

Reviewer #3 (Remarks to the Author):

I appreciate the authors for their detailed reply.

I am confused with the double headed arrows in Bloch sphere in Supplementary Figure 2 e-f which are not well defined (usually pseudospin is represented as single headed arrow). Let me interpret these as pairs of eigenstates, which are energetically degenerate (otherwise there should be time reversal symmetry breaking due to the finite $|X_{A,K}\rangle$ population which has non-zero angular momentum). Since all pseudospin states in the Bloch sphere can be represented with the superposition of these paired eigenstates which are energetically degenerate, that means there is essentially no valley mixing (all pseudospin states are energetically degenerate). I imagine that the authors solved the problem numerically and obtained these pairs of eigenstates with random orientation in Supplementary Figure 2e-f. I suspect that the numerical approach which the authors took hinders the essential understanding of the problem. I rather recommend to solve the problem analytically with some approximation to clarify the symmetry of the problem (For example, what determines the axis of the pseudospin states in Supplementary Fig. 2e-f?)

I trust the experimental results with considerable efforts and the paper is worth to be published. However, I afraid that this unclear discussion which would mislead the community in future. Compared to the experimental efforts which the authors made, solving this problem analytically is not a big deal and I hope that the authors convince the readers properly with bit more efforts.

Response: We thank the reviewer for the critical question about the valley pseudospin vectors. We indeed used the doubled headed arrow to represent a pair of degenerate eigenstates which are time-reversal counterparts. We agree with the reviewer that all pseudospin states in the Bloch sphere are degenerate eigenstates and the pseudospin vectors in Fig. 2e-f were indeed randomly chosen in the numerical simulation. Therefore, there is essentially no valley mixing as pointed out by the reviewer. To address the issue, we follow the reviewer's suggestion to solve the moiré exciton states analytically by treating the moiré potential as a perturbation. Our analytical results reveal the nature of moiré excitons in 1L/1L WSe₂/WS₂. Furthermore, we consider the intralayer-

like and interlayer-like hybrid excitons in 2L WSe₂ due to the interlayer tunneling between valence bands. We show that the intralayer-like hybrid excitons coupled to the moiré potential can give rise to the moiré excitons in 2L/1L WSe₂/WS₂. To understand the energy shifts of moiré excitons in 2L/1L WSe₂/WS₂ compared with those in 1L/1L WSe₂/WS₂, we propose a possible mechanism by considering the hybridization between moiré excitons and interlayer-like hybrid excitons. The similar analysis can be extended to 3L/1L WSe₂/WS₂. The exciton hybridization picture provides a possible microscopic mechanism responsible for the phenomenological model (Eqn. R10) that well explains our experimental observations, while the exact mechanism warrants future exploration.

The detailed analytical calculation of moiré excitons and extensive discussion about hybrid excitons are provided in the reply below and updated in the Supplementary Information. The figures of pseudospin vectors are eliminated. The manuscript has also been revised accordingly.

Nature of moiré excitons in 1L/1L WSe₂/WS₂

To reveal the nature of moiré excitons, we first consider the WSe₂ intralayer A exciton described by the effective Hamiltonian

$$H_{A,\mathbf{Q}} = \left(E_0 + \frac{\hbar^2 \mathbf{Q}^2}{2M} \right) \tau_0 + J|\mathbf{Q}| \tau_x + J|\mathbf{Q}| [\cos(2\theta_{\mathbf{Q}}) \tau_x + \sin(2\theta_{\mathbf{Q}}) \tau_y]$$

where τ_0 and $\tau_{x,y}$ are the identity matrix and Pauli matrices acting on the valley space [1,2]. \mathbf{Q} is the exciton center-of-mass momentum whose polar angle is $\theta_{\mathbf{Q}}$ and $M = 0.64m_e$ is the total mass of an electron-hole pair in WSe₂ [3]. The eigenstates and eigenvalues of H_A are:

$$|\psi_{\pm,\mathbf{Q}}\rangle = \frac{1}{\sqrt{2}} \begin{pmatrix} e^{-i\theta_{\mathbf{Q}}} \\ \pm e^{i\theta_{\mathbf{Q}}} \end{pmatrix}, \quad E_{\pm,\mathbf{Q}} = E_0 + \frac{\hbar^2 \mathbf{Q}^2}{2M} + J|\mathbf{Q}| \pm J|\mathbf{Q}|$$

Note that the two eigenstates are degenerate at $\mathbf{Q} = \mathbf{0}$ and any superposition of $|\psi_{\pm,0}\rangle$ remains the eigenstate of $H_{A,0}$. Namely, there is an emergent valley pseudospin rotational symmetry, i.e., $H_{A,0}$ is invariant under any pseudospin rotation operation. However, this emergent symmetry is merely induced by combining the two A exciton states from opposite valleys together in $H_{A,\mathbf{Q}}$ even when they are not hybridized at $\mathbf{Q} = \mathbf{0}$. Since valley is a good quantum number for the bright A exciton states, we should fix

$$|\psi_{+,0}\rangle = |X_{A,K}\rangle = \begin{pmatrix} 1 \\ 0 \end{pmatrix}, \quad |\psi_{-,0}\rangle = |X_{A,K'}\rangle = \begin{pmatrix} 0 \\ 1 \end{pmatrix}$$

For $\mathbf{Q} \neq \mathbf{0}$, the nonzero intervalley exchange interaction hybridizes $|X_{A,K}\rangle$ and $|X_{A,K'}\rangle$, and breaks the pseudospin rotational symmetry.

In 1L/1L WSe₂/WS₂, the exciton moiré potential reads

$$V_X(\mathbf{r}) = \sum_{i=1}^6 V_i \exp(i\mathbf{G}_i \cdot \mathbf{r})$$

where $V_{1,3,5} = V \exp(i\phi)$, $V_{2,4,6} = V \exp(-i\phi)$, and $\mathbf{G}_{1,3,5} = -\mathbf{G}_{2,4,6}$ (see Fig. R1a) [4]. $J = 0.04$ eV·nm, $V = 25$ meV and $\phi = 15^\circ$ [5]. Then the moiré exciton can be described by the Hamiltonian

$$H_{MX} = H_{A,Q} + V_X(\mathbf{r}) \quad (\text{R1})$$

The moiré potential can couple two excitonic states if their momenta are differed by a primitive moiré reciprocal lattice vector \mathbf{G}_i with $i \in \{1, \dots, 6\}$. Therefore, the bright A exciton states $|X_{A,K}\rangle$ and $|X_{A,K'}\rangle$ (at γ in Fig. R1a) can couple to the twelve Umklapp states $\{|\psi_{\pm, \mathbf{G}_i}\rangle\}$ in the first shell as

$$\langle \psi_{\pm, \mathbf{G}_i} | V_X | X_{A,K} \rangle = \frac{V_i}{\sqrt{2}} e^{i\theta_{\mathbf{G}_i}}, \quad \langle \psi_{\pm, \mathbf{G}_i} | V_X | X_{A,K'} \rangle = \pm \frac{V_i}{\sqrt{2}} e^{-i\theta_{\mathbf{G}_i}}$$

Moreover, the twelve Umklapp states can also mutually couple with their nearest neighbors as

$$\langle \psi_{\pm, \mathbf{G}_{i+1}} | V_X | \psi_{\pm, \mathbf{G}_i} \rangle = \frac{V_{i+2}}{2}, \quad \langle \psi_{\pm, \mathbf{G}_{i+1}} | V_X | \psi_{\mp, \mathbf{G}_i} \rangle = \frac{i\sqrt{3}V_{i+2}}{2}$$

where we use the fact that $\mathbf{G}_{i+1} - \mathbf{G}_i = \mathbf{G}_{i+2}$ and $\theta_{\mathbf{G}_{i+1}} - \theta_{\mathbf{G}_i} = \pi/3$. Here we consider \mathbf{G}_i for $i \in \{1, \dots, 6\}$ forming a loop, i.e., $\mathbf{G}_{6+n} = \mathbf{G}_n$ and $V_{6+n} = V_n$.

Fig. R1. (a) Schematic Umklapp scattering induced by the moiré potential. The black hexagons are the MBZs. (b-d) Exciton minibands of 1L/1L WSe₂/WS₂ with $V = 0, 5, 25$ meV. The red dashed curves represent the bare dispersion of WSe₂ intralayer A exciton. The energy bands are plotted along the green dashed path in (a).

Now we treat the moiré potential as a perturbation and consider the leading order corrections to the bright A exciton states as

$$\begin{aligned}
|X_{I,+}^{1L}\rangle &= |X_{A,K}\rangle + \sum_{\xi=\pm} \sum_{i=1}^6 \frac{\langle \psi_{\xi,G_i} | V_X | X_{A,K} \rangle}{E_0 - E_{\xi,G_i}} |\psi_{\xi,G_i}\rangle \\
&= |X_{A,K}\rangle - \sum_{i=1}^6 \frac{V_i e^{i\theta_{G_i}}}{\sqrt{2}} \left(\frac{|\psi_{+,G_i}\rangle}{\frac{\hbar^2 b_M^2}{2M} + 2Jb_M} + \frac{|\psi_{-,G_i}\rangle}{\frac{\hbar^2 b_M^2}{2M}} \right) \quad (R2)
\end{aligned}$$

$$\begin{aligned}
|X_{I,-}^{1L}\rangle &= |X_{A,K'}\rangle + \sum_{\xi=\pm} \sum_{i=1}^6 \frac{\langle \psi_{\xi,G_i} | V_X | X_{A,K'} \rangle}{E_0 - E_{\xi,G_i}} |\psi_{\xi,G_i}\rangle \\
&= |X_{A,K'}\rangle - \sum_{i=1}^6 \frac{V_i e^{-i\theta_{G_i}}}{\sqrt{2}} \left(\frac{|\psi_{+,G_i}\rangle}{\frac{\hbar^2 b_M^2}{2M} + 2Jb_M} - \frac{|\psi_{-,G_i}\rangle}{\frac{\hbar^2 b_M^2}{2M}} \right) \quad (R3)
\end{aligned}$$

where $b_M = |\mathbf{G}_i|$. In principle, the two degenerate bright A exciton states from opposite valleys can hybridize through the second order process mediated by the coupling with $|\psi_{\pm,G_i}\rangle$. According to the degenerate second order perturbation theory, the effective coupling between $|X_{A,K}\rangle$ and $|X_{A,K'}\rangle$ by integrating out the Umklapp states is

$$\begin{aligned}
T_{K,K'} &= \sum_{\xi=\pm} \sum_{i=1}^6 \frac{\langle X_{A,K} | V_X | \psi_{\xi,G_i} \rangle \langle \psi_{\xi,G_i} | V_X | X_{A,K'} \rangle}{E_0 - E_{\xi,G_i}} = \sum_{i=1}^6 \frac{V^2}{2} e^{-2i\theta_{G_i}} \left(\frac{2M}{\hbar^2 b_M^2} - \frac{2M}{\hbar^2 b_M^2 + 4Jb_M M} \right) \\
&= 0 \quad (R4)
\end{aligned}$$

The effective intervalley coupling vanishes due to the quantum interference since $\sum_{i=1}^6 e^{-2i\theta_{G_i}} = 0$. Therefore, the two moiré excitons $X_{I,\pm}^{1L}$ inherit components from the Umklapp states and remain degenerate at

$$\begin{aligned}
E_{X_{I,+}^{1L}} &= E_0 + \sum_{\xi=\pm} \sum_{i=1}^6 \frac{|\langle \psi_{\xi,G_i} | V_X | X_{A,K} \rangle|^2}{E_0 - E_{\xi,G_i}} = E_0 - \frac{6V^2 M}{\hbar^2 b_M^2 + 4Jb_M M} - \frac{6V^2 M}{\hbar^2 b_M^2} \\
E_{X_{I,-}^{1L}} &= E_0 + \sum_{\xi=\pm} \sum_{i=1}^6 \frac{|\langle \psi_{\xi,G_i} | V_X | X_{A,K'} \rangle|^2}{E_0 - E_{\xi,G_i}} = E_{X_{I,+}^{1L}}
\end{aligned}$$

There is no hybridization between $|X_{A,K}\rangle$ and $|X_{A,K'}\rangle$ which would otherwise break the degeneracy.

Fig. R2. (a) Schematic optical interband transitions and the formation of A and B excitons in monolayer WSe₂. The red (blue) energy levels denote the electronic states with spin up (down) at the *K* valley. (b) The formation of moiré excitons in 1L/1L WSe₂/WS₂ through the hybridization between WSe₂ A exciton states and Umklapp states. The yellow and black energy levels are for the bright and dark excitonic states. The dashed lines denote the hybridization between different excitonic states.

The Umklapp states can also inherit components from the bright A exciton states and generate additional bright moiré excitons (see Fig. R2). To show this explicitly, we consider the lowest six degenerate Umklapp states $\{|\psi_{-G_i}\rangle\}$. The leading order correction comes from the mutual coupling among themselves. According to the degenerate perturbation theory, diagonalizing the 6×6 matrix whose nonzero elements are

$$\langle \psi_{-G_{i+1}} | V_X | \psi_{-G_i} \rangle = \frac{V_{i+2}}{2}$$

yields the six perturbed Umklapp states

$$\begin{aligned}
 |\psi_1\rangle &= \frac{1}{\sqrt{6}} \left(\sum_{i=1,3,5} e^{-\frac{i\phi}{2}} |\psi_{-G_i}\rangle - \sum_{i=2,4,5} e^{\frac{i\phi}{2}} |\psi_{-G_i}\rangle \right) \\
 |\psi_2\rangle &= \frac{1}{2} \left(e^{-\frac{i\phi}{2}} |\psi_{-G_1}\rangle - e^{\frac{i\phi}{2}} |\psi_{-G_2}\rangle + e^{\frac{i\phi}{2}} |\psi_{-G_4}\rangle - e^{-\frac{i\phi}{2}} |\psi_{-G_5}\rangle \right) \\
 |\psi_3\rangle &= \frac{1}{2\sqrt{3}} \left(e^{-\frac{i\phi}{2}} |\psi_{-G_1}\rangle + e^{\frac{i\phi}{2}} |\psi_{-G_2}\rangle - 2e^{-\frac{i\phi}{2}} |\psi_{-G_3}\rangle + e^{\frac{i\phi}{2}} |\psi_{-G_4}\rangle + e^{-\frac{i\phi}{2}} |\psi_{-G_5}\rangle \right. \\
 &\quad \left. - 2e^{\frac{i\phi}{2}} |\psi_{-G_6}\rangle \right) \\
 |\psi_4\rangle &= \frac{1}{2} \left(e^{-\frac{i\phi}{2}} |\psi_{-G_1}\rangle + e^{\frac{i\phi}{2}} |\psi_{-G_2}\rangle - e^{\frac{i\phi}{2}} |\psi_{-G_4}\rangle - e^{-\frac{i\phi}{2}} |\psi_{-G_5}\rangle \right) \\
 |\psi_5\rangle &= \frac{1}{2\sqrt{3}} \left(e^{-\frac{i\phi}{2}} |\psi_{-G_1}\rangle - e^{\frac{i\phi}{2}} |\psi_{-G_2}\rangle - 2e^{-\frac{i\phi}{2}} |\psi_{-G_3}\rangle - e^{\frac{i\phi}{2}} |\psi_{-G_4}\rangle + e^{-\frac{i\phi}{2}} |\psi_{-G_5}\rangle \right. \\
 &\quad \left. + 2e^{\frac{i\phi}{2}} |\psi_{-G_6}\rangle \right) \\
 |\psi_6\rangle &= \frac{1}{\sqrt{6}} \left(\sum_{i=1,3,5} e^{-\frac{i\phi}{2}} |\psi_{-G_i}\rangle + \sum_{i=2,4,5} e^{\frac{i\phi}{2}} |\psi_{-G_i}\rangle \right)
 \end{aligned}$$

Here $|\psi_1\rangle$ and $|\psi_6\rangle$ are nondegenerate with $E_1 = E_0 + \frac{\hbar^2 b_M^2}{2m} - V$ and $E_6 = E_0 + \frac{\hbar^2 b_M^2}{2m} + V$, while $|\psi_{2,3}\rangle$ and $|\psi_{4,5}\rangle$ are both double degenerate with $E_{2,3} = E_0 + \frac{\hbar^2 b_M^2}{2M} - \frac{V}{2}$ and $E_{4,5} = E_0 + \frac{\hbar^2 b_M^2}{2M} + \frac{V}{2}$. Furthermore, we consider the first-order correction to $\{|\psi_i\rangle\}$ from their coupling with $|X_{A,K}\rangle$ and $|X_{A,K'}\rangle$ as

$$|\Psi_i\rangle = |\psi_i\rangle + \sum_{\tau=K,K'} \frac{\langle X_{A,\tau} | V_X | \psi_i \rangle}{E_i - E_0} |X_{A,\tau}\rangle$$

that yields

$$|\Psi_1\rangle = |\psi_1\rangle$$

$$|\Psi_2\rangle = |\psi_2\rangle - \frac{(\sqrt{3} + 3i)V \sin \frac{3\phi}{2}}{2\sqrt{2} \left(\frac{\hbar^2 b_M^2}{2M} - \frac{V}{2} \right)} |X_{A,K}\rangle - \frac{(\sqrt{3} - 3i)V \sin \frac{3\phi}{2}}{2\sqrt{2} \left(\frac{\hbar^2 b_M^2}{2M} - \frac{V}{2} \right)} |X_{A,K'}\rangle \quad (\text{R5})$$

$$|\Psi_3\rangle = |\psi_3\rangle + \frac{(3 - \sqrt{3}i)V \sin \frac{3\phi}{2}}{2\sqrt{2} \left(\frac{\hbar^2 b_M^2}{2M} - \frac{V}{2} \right)} |X_{A,K}\rangle + \frac{(3 + \sqrt{3}i)V \sin \frac{3\phi}{2}}{2\sqrt{2} \left(\frac{\hbar^2 b_M^2}{2M} - \frac{V}{2} \right)} |X_{A,K'}\rangle \quad (\text{R6})$$

$$|\Psi_4\rangle = |\psi_4\rangle + \frac{(3 - \sqrt{3}i)V \cos \frac{3\phi}{2}}{2\sqrt{2} \left(\frac{\hbar^2 b_M^2}{2M} + \frac{V}{2} \right)} |X_{A,K}\rangle - \frac{(3 + \sqrt{3}i)V \cos \frac{3\phi}{2}}{2\sqrt{2} \left(\frac{\hbar^2 b_M^2}{2M} + \frac{V}{2} \right)} |X_{A,K'}\rangle \quad (\text{R7})$$

$$|\Psi_5\rangle = |\psi_5\rangle + \frac{(\sqrt{3} + 3i)V \cos \frac{3\phi}{2}}{2\sqrt{2} \left(\frac{\hbar^2 b_M^2}{2M} + \frac{V}{2} \right)} |X_{A,K}\rangle - \frac{(\sqrt{3} - 3i)V \cos \frac{3\phi}{2}}{2\sqrt{2} \left(\frac{\hbar^2 b_M^2}{2M} + \frac{V}{2} \right)} |X_{A,K'}\rangle \quad (\text{R8})$$

$$|\Psi_6\rangle = |\psi_6\rangle$$

Because $|\Psi_{2,3}\rangle$ and $|\Psi_{4,5}\rangle$ inherit components from the bright A excitons, they become the bright moiré excitons as

$$|X_{II,+}^{1L}\rangle = |\Psi_2\rangle, \quad |X_{II,-}^{1L}\rangle = |\Psi_3\rangle, \quad |X_{III,+}^{1L}\rangle = |\Psi_4\rangle, \quad |X_{III,-}^{1L}\rangle = |\Psi_5\rangle \quad (\text{R9})$$

where $|X_{II,\pm}^{1L}\rangle$ and $|X_{III,\pm}^{1L}\rangle$ are both double degenerate as

$$E_{X_{II,+}^{1L}} = E_2 + \sum_{\tau=K,K'} \frac{|\langle X_{A,\tau} | V_X | X_{II,+}^{1L} \rangle|^2}{E_2 - E_0} = E_0 + \frac{\hbar^2 b_M^2}{2M} - \frac{V}{2} + \frac{3V^2 \sin^2 \frac{3\phi}{2}}{\frac{\hbar^2 b_M^2}{2M} - \frac{V}{2}}$$

$$E_{X_{II,-}^{1L}} = E_3 + \sum_{\tau=K,K'} \frac{|\langle X_{A,\tau} | V_X | X_{II,-}^{1L} \rangle|^2}{E_3 - E_0} = E_{X_{II,+}^{1L}}$$

$$E_{X_{III,+}^{1L}} = E_4 + \sum_{\tau=K,K'} \frac{|\langle X_{A,\tau} | V_X | X_{III,+}^{1L} \rangle|^2}{E_4 - E_0} = E_0 + \frac{\hbar^2 b_M^2}{2M} + \frac{V}{2} + \frac{3V^2 \cos^2 \frac{3\phi}{2}}{\frac{\hbar^2 b_M^2}{2M} + \frac{V}{2}}$$

$$E_{X_{III,-}^{1L}} = E_5 + \sum_{\tau=K,K'} \frac{|\langle X_{A,\tau} | V_X | X_{III,-}^{1L} \rangle|^2}{E_5 - E_0} = E_{X_{III,+}^{1L}}$$

Therefore, any superposition of $|X_{II,\pm}^{1L}\rangle$ or $|X_{III,\pm}^{1L}\rangle$ are still the bright moiré exciton states. The nondegenerate $|\Psi_1\rangle$ and $|\Psi_6\rangle$ are orthogonal to the bright A exciton states and remain dark (see Fig. R2b). These results are consistent with our numerical simulation of Eq. (R1) by the plane wave expansion method, as shown in Fig. R1b-d. There is another contribution to the first order correction of $\{|\psi_i\rangle\}$ from their coupling with $\{\psi_{+,G_i}\}$, which does not affect the brightness of the moiré excitons and is omitted for simplification.

Hybrid excitons in 2L WSe₂

To understand the moiré excitons in 2L/1L WSe₂/WS₂, we first consider the interlayer hybridization in 2L WSe₂. Due to the AB-stacking, the 1st WSe₂ layer is rotated by 180° with respect to the 2nd WSe₂ layer. Therefore, the energy bands of the two layers have opposite spin polarization in the same valley. At the K and K' points, the spin-conserved interlayer tunneling is forbidden for conduction band electrons due to the symmetry constrain, while it is allowed for valence band holes (see Fig. R3a) [6]. The hybridization between two valence bands (with same spin) in different layers can be described by the Hamiltonian $\begin{pmatrix} \lambda & t_{\perp} \\ t_{\perp} & -\lambda \end{pmatrix}$ where $2\lambda = 0.456$ eV is the valence band splitting in 1L WSe₂ and $t_{\perp} = 0.067$ eV is the interlayer hopping [6]. The interlayer hybridization enlarges the valence band splitting to $2\sqrt{\lambda^2 + t_{\perp}^2}$ and redistributes the valence band holes whose wave function has a small portion of $\sim \frac{t_{\perp}}{t_{\perp} + 2\lambda} = 12.8\%$ in the other layer.

Fig. R3. (a) Schematic optical interband transitions and the formation of hybrid excitons at the K valley of 2L WSe₂. (b) The spatial charge distribution of intralayer-like and interlayer-like hybrid excitons in 2L WSe₂. (c) The formation of moiré excitons in 2L/1L WSe₂/WS₂ through the hybridization between 2L WSe₂ intralayer-like exciton states and Umklapp states. (d) The energy shifts of moiré excitons through their hybridization with 2L WSe₂ interlayer-like exciton states in 2L/1L WSe₂/WS₂. The yellow and black solid energy levels are for the bright and dark excitonic states. The yellow dashed energy levels denote the intralayer-like exciton states with very weak oscillator strength. The dashed lines indicate the hybridization between different excitonic states.

The hybrid valence bands enable four possible optical interband transitions in the K valley, as shown in Fig. R3a, and their time-reversal counterparts in the K' valley [7]. Here the predominate interband transitions (marked by yellow solid arrows) lead to the degenerate intralayer-like hybrid excitons $hX_{K,\uparrow}$ and $hX_{K,\downarrow}$, while the subordinate interband transitions (marked by yellow dashed arrows) generate the degenerate interlayer-like hybrid excitons $iX_{K,\uparrow}$ and $iX_{K,\downarrow}$, as shown in Fig. R3b. The hybrid excitons $hX_{K',\sigma}$ and $iX_{K',\sigma}$ (where the spin index $\sigma = \uparrow, \downarrow$) in the K' valley can be obtained by the time-reversal symmetry. The oscillator strength of interlayer-like excitons $iX_{\tau,\sigma}$ (where the valley index $\tau = K, K'$) is much weaker than that of intralayer-like excitons $hX_{\tau,\sigma}$. The energy difference between $hX_{\tau,\sigma}$ and $iX_{\tau,\sigma}$ depends on the conduction band splitting (see Fig. R3a) which is about 37 meV in WSe₂ [3]. Because the binding energy of $iX_{\tau,\sigma}$ is smaller than that of $hX_{\tau,\sigma}$ due to the larger electron-hole separation, the energy

difference between $hX_{\tau,\sigma}$ and $iX_{\tau,\sigma}$ is smaller than the conduction band splitting and cannot be resolved in the experiment (see Fig. R4). It naturally explains why the resonant peak in 1L WSe₂ is sharper than that in 2L WSe₂. Furthermore, there is a redshift of the resonant peak in 2L WSe₂ compared with that in 1L WSe₂ by about 30 meV. This could be due to the change of dielectric environment as well as the reduction of energy gap in 2L WSe₂ (see Fig. R3a). Therefore, the hybrid excitons $hX_{\tau,\sigma}$ and $iX_{\tau,\sigma}$ in 2L WSe₂ have lower energy than the intralayer A excitons $X_{A,\tau}$ in 1L WSe₂.

Fig. R4. Differential reflectance spectra of 1L, 2L, 3L WSe₂, and 2L/1L WSe₂/WS₂.

Possible mechanism for moiré excitons in 2L/1L and 3L/1L WSe₂/WS₂

Now we consider the moiré excitons in 2L/1L WSe₂/WS₂. For the predominant intralayer-like hybrid excitons $\{hX_{\tau,\sigma}\}$, the moiré potential will affect $hX_{K,\uparrow}$ and $hX_{K',\downarrow}$ which are mainly distributed in the 1st WSe₂ layer interfacing WS₂, while $hX_{K,\downarrow}$ and $hX_{K',\uparrow}$ highly localized in the 2nd WSe₂ layer are expected to be unaffected (see Fig. R3b). According to the same mechanism in 1L/1L WSe₂/WS₂, the moiré potential will hybridize $|hX_{K,\uparrow}\rangle$ and $|hX_{K',\downarrow}\rangle$ with the Umklapp states $\{|\psi_{\xi,G_i}^{2L}\rangle\}$ and lead to three double degenerate moiré excitons $|X_{\mu,\pm}^{2L}\rangle$ (where $\mu = I, II, III$) through the Umklapp scattering in 2L/1L WSe₂/WS₂ (see Fig. R3c). The moiré exciton states $|X_{\mu,\pm}^{2L}\rangle$ should be in the similar form as $|X_{\mu,\pm}^{1L}\rangle$ in Eq. (R2,R3,R5-R9) but with the 1L WSe₂ $|X_{A,K}\rangle$, $|X_{A,K'}\rangle$, $|\psi_{\xi,G_i}\rangle$ replaced by the 2L WSe₂ $|hX_{K,\uparrow}\rangle$, $|hX_{K',\downarrow}\rangle$, $|\psi_{\xi,G_i}^{2L}\rangle$. The remaining $|hX_{K,\downarrow}\rangle$ and $|hX_{K',\uparrow}\rangle$ in the 2nd WSe₂ layer give rise to the $|X_{IV,\pm}^{2L}\rangle$ whose resonant energy is coincident with that in 2L WSe₂ (see Fig. R4). Therefore, the four resonant peaks observed in 2L/1L WSe₂/WS₂ can be essentially explained by considering the intralayer-like hybrid excitons in 2L WSe₂ coupled to the moiré potential.

Another important feature of the moiré excitons observed in 2L/1L WSe₂/WS₂ is that there is a redshift in $X_{I,\pm}^{2L}$ but blueshifts in $X_{II,\pm}^{2L}$ and $X_{III,\pm}^{2L}$ compared with those in 1L/1L WSe₂/WS₂ (see Fig. 1b of the main text). To understand the energy shifts of moiré excitons, we propose a possible mechanism by further considering the interlayer-like hybrid excitons $\{iX_{\tau,\sigma}\}$. It is noted that $hX_{\tau,\sigma}$

and $iX_{\tau,\sigma}$ [with $(\tau, \sigma) = (K, \uparrow)$ or (K', \downarrow)] share the same valence band holes mainly in the 1st WSe₂ layer but have different conduction band electrons in the 1st and 2nd WSe₂ layers, respectively (see Fig. R3b). Although the interlayer tunneling between conduction bands is hindered by the symmetry at K and K' points [6], it can be nonzero away from these high-symmetry points. Therefore, the Umklapp states $\{|\psi_{\xi, G_i}^{2L}\rangle\}$ in 2L WSe₂ should be formed by electrons and holes (with finite momenta away from K and K') in the hybrid conduction and valence bands, respectively. Meanwhile, the interlayer-like excitons $iX_{\tau,\sigma}$ is about half distributed in the 1st WSe₂ layer and should also feel a moiré potential $V'_X(\mathbf{r})$ that leads to the coupling with $|\psi_{\xi, G_i}^{2L}\rangle$. This paves the way for the hybridization between $|X_{\mu, \pm}^{2L}\rangle$ and $|iX_{\tau,\sigma}\rangle$ with the coupling constant $t_\mu = \langle iX_{\tau,\sigma} | V'_X(\mathbf{r}) | X_{\mu, \pm}^{2L} \rangle$ since $|X_{\mu, \pm}^{2L}\rangle$ contain the components of $|\psi_{\xi, G_i}^{2L}\rangle$. Furthermore, the Umklapp components are dominate in $|X_{II, \pm}^{2L}\rangle$ and $|X_{III, \pm}^{2L}\rangle$ that can result in stronger hybridization with $|iX_{\tau,\sigma}\rangle$. Then this hybridization is expected to induce the energy shifts of moiré excitons, as sketched in Fig. R3d. Note that $|X_{IV, \pm}^{2L}\rangle$ highly localized in the second WSe₂ layer is barely affected by the hybridization. The similar analysis can be extended to 3L/1L WSe₂/WS₂ in which additional hybrid excitons can be induced by the interlayer tunneling between valence bands in the 2nd and 3rd WSe₂ layers. In this case, the additional hybrid excitons away from the WSe₂/WS₂ interface do not affect the moiré excitons. Therefore, the moiré excitons $X_{\mu, \pm}^{3L}$ in 3L/1L WSe₂/WS₂ are nearly identical to $X_{\mu, \pm}^{2L}$ in 2L/1L WSe₂/WS₂, as shown in Fig. 1b of the main text.

To evaluate $t_\mu = \langle iX_{\tau,\sigma} | V'_X(\mathbf{r}) | X_{\mu, \pm}^{nL} \rangle$ (where $n = 2$ or 3), one needs to know the exact hybrid exciton states in nL WSe₂ and the moiré potentials for different hybrid excitons in $nL/1L$ WSe₂/WS₂, which to our knowledge remains unknown and is beyond the scope of this work. Further DFT studies can be stimulated by our experimental results. Nevertheless, we expect the essential physics associated with the energy shifts of moiré excitons can be captured by the 4×4 phenomenological model Hamiltonian

$$H = \begin{pmatrix} E_{iX} & t_I & t_{II} & t_{III} \\ t_I^* & E_I & 0 & 0 \\ t_{II}^* & 0 & E_{II} & 0 \\ t_{III}^* & 0 & 0 & E_{III} \end{pmatrix} \quad (\text{R10})$$

in the basis of $\{|iX\rangle, |X_I^{nL}\rangle, |X_{II}^{nL}\rangle, |X_{III}^{nL}\rangle\}$. Here we omit the subscript for simplification because each moiré exciton is double degenerate due to the time-reversal symmetry. Because the interlayer-like hybrid exciton iX has very weak oscillator strength and cannot be resolved in the experiment, we approximately treat it as a dark exciton. Diagonalizing the model Hamiltonian Eq. (R10) yields the hybrid moiré excitons with enlarged energy separation. The magnitude of t_μ can be estimated from the energy shifts of moiré exciton resonances. To show the optical response of the moiré system, we calculate the real part of the optical conductivity that gives the optical absorption [4]. According to the experimental data, we take $E_I = 1.662$ eV, $E_{II} = 1.715$ eV, and $E_{III} = 1.753$ eV. We approximate the energy of iX by $E_{iX} = 1.700$ eV since it cannot be resolved in the experiment. $t_I = -0.03$ eV, $t_{II} = -0.04$ eV, and $t_{III} = 0.04i$ eV are chosen to match the experimental results in Fig. 1b. The numerical results are shown in Fig. 2c-e for 1L/1L, 2L/1L,

and 3L/1L WSe₂/WS₂, respectively. Here we also consider the bright hybrid excitons highly in the upper WSe₂ layers that give rise to $X_{IV}^{2L/3L}$. In particular, the two sub-resonances around $E = 1.69$ eV in Fig. 2e are consistent with the broad resonant peak X_{IV}^{3L} in Fig. 1b which splits into the two sub-resonances in the p-doped site in Fig. 1e.

References

1. Yu, H., Liu, G.-B., Gong, P., Xu, X. & Yao, W. Dirac cones and Dirac saddle points of bright excitons in monolayer transition metal dichalcogenides. *Nature Commun.* 5, 3876 (2014).
2. Wu, F., Qu, F. & MacDonald, A. H. Exciton band structure of monolayer MoS₂. *Phys. Rev. B* 91, 075310 (2015).
3. Kormányos, A. et al. k·p theory for two-dimensional transition metal dichalcogenide semiconductors. *2D Mater.* 2, 022001 (2015).
4. Wu, F., Lovorn, T. & MacDonald, A. H. Topological exciton bands in moiré heterojunctions. *Phys. Rev. Lett.* 118, 147401 (2017).
5. Jin, C. H. et al. Observation of moiré excitons in WSe₂/WS₂ heterostructure superlattices. *Nature* 567, 76–80 (2019).
6. Gong, Z. et al. Magnetoelectric effects and valley-controlled spin quantum gates in transition metal dichalcogenide bilayers. *Nature Commun.* 4, 15 (2013).
7. Hsu, W. et al. Tailoring excitonic states of van der Waals bilayers through stacking configuration, band alignment, and valley spin. *Sci. Adv.* 5, eaax7407 (2019).

REVIEWERS' COMMENTS

Reviewer #3 (Remarks to the Author):

I thank the authors for their efforts to figure out the microscopic physics behind their observations. To couple $|iX_K, \uparrow\rangle$ with $|\psi^{2L_\xi, Gi}\rangle$, the interlayer tunnel coupling of conduction band states is also required. Given that the interlayer tunneling of the conduction band state at K point is prohibited by rotational symmetry in 2H stacked bilayer TMDs, I am not sure this is plausible. However, I cannot deny the possibility that some Umklapp scattered conduction band state may carry proper angular momentum for the interlayer tunnel coupling. The origin of X_{IV} state is supported without this argument of this interlayer exciton coupling, so it may not be critical for the qualitative interpretation of the data. Again, I thank the authors for their efforts to deepen the discussion. I think the paper is ready for publication.

(Line 98, Line 103 of supplementary, $i = 2,4,5$ is probably mistype of $i = 2,4,6$ I think.)